# Post-processing for Individual Fairness

**Felix Petersen**[*]
University of Konstanz
felix.petersen@uni.kn

**Debarghya Mukherjee**[*]
University of Michigan
mdeb@umich.edu

**Yuekai Sun**
University of Michigan
yuekai@umich.edu

**Mikhail Yurochkin**
IBM Research, MIT-IBM Watson AI Lab
mikhail.yurochkin@ibm.com

## Abstract

Post-processing in algorithmic fairness is a versatile approach for correcting bias in ML systems that are already used in production. The main appeal of post-processing is that it avoids expensive retraining. In this work, we propose general post-processing algorithms for individual fairness (IF). We consider a setting where the learner only has access to the predictions of the original model and a similarity graph between individuals guiding the desired fairness constraints. We cast the IF post-processing problem as a graph smoothing problem corresponding to graph Laplacian regularization that preserves the desired "treat similar individuals similarly" interpretation. Our theoretical results demonstrate the connection of the new objective function to a local relaxation of the original individual fairness. Empirically, our post-processing algorithms correct individual biases in large-scale NLP models such as BERT, while preserving accuracy.

## 1   Introduction

There are many instances of algorithmic bias in machine learning (ML) models [1]–[4], which has led to the development of methods for *quantifying* and *correcting* algorithmic bias. To quantify algorithmic bias, researchers have proposed numerous mathematical definitions of algorithmic fairness. Broadly speaking, these definitions fall into two categories: *group fairness* [5] and *individual fairness* [6]. The former formalizes the idea that ML system should treat certain *groups* of individuals similarly, e.g., requiring the average loan approval rate for applicants of different ethnicities be similar [7]. The latter asks for similar treatment of similar *individuals*, e.g., same outcome for applicants with resumes that differ only in names [8]. Researchers have also developed many ways of correcting algorithmic bias. These fairness interventions broadly fall into three categories: pre-processing the data, enforcing fairness during model training (also known as in-processing), and post-processing the outputs of a model.

While both group and individual fairness (IF) definitions have their benefits and drawbacks [5], [6], [9], the existing suite of algorithmic fairness solutions mostly enforces group fairness. The few prior works on individual fairness are all in-processing methods [10]–[13]. Although in-processing is arguably the most-effective type of intervention, it has many practical limitations. For example, it requires training models from scratch. Nowadays, it is more common to fine-tune publicly available models (e.g., language models such as BERT [14] and GPT-3 [15]) than to train models afresh, as many practitioners do not have the necessary computational resources. Even with enough computational resources, training large deep learning models has a significant environmental impact [4], [16].

---

[*]Equal Contribution.

35th Conference on Neural Information Processing Systems (NeurIPS 2021).

Post-processing offers an easier path towards incorporating algorithmic fairness into deployed ML models, and has potential to reduce environmental harm from re-training with in-processing fairness techniques.

In this paper, we propose a computationally efficient method for post-processing off-the-shelf models to be *individually fair*. We consider a setting where we are given the outputs of a (possibly unfair) ML model on a set of $n$ individuals, and side information about their similarity for the ML task at hand, which can either be obtained using a fair metric on the input space or from external (e.g., human) annotations. Our starting point is a post-processing version of the algorithm by Dwork *et al.* [6] (see (2.3)). Unfortunately, this method has two drawbacks: poor scalability and an unfavorable trade-off with accuracy. As we shall see, the sharp trade-off is due to the restrictions imposed on dissimilar individuals by Dwork *et al.* [6]'s global Lipschitz continuity condition. By relaxing these restrictions on dissimilar individuals, we obtain a better trade-off between accuracy and fairness, while preserving the intuition of treating *similar* individuals similarly. This leads us to consider a graph signal-processing approach to IF post-processing that only enforces similar outputs between similar individuals. The nodes in the underlying graph correspond to individuals, edges (possibly weighted) indicate similarity, and the signal on the graph is the output of the model on the corresponding node-individuals. To enforce IF, we use Laplacian regularization [17], which encourages the signal to be smooth on the graph. We illustrate this idea in Figure 1: a biased model decides whom to show a job ad for a Python programming job based on their CVs and chooses Charlie and Dave but excludes Alice. However, from the qualifications, we can see that Alice, Charlie and Dave are similar because they all have experience in Python, which is the job requirement, and thus should be treated similarly. We represent all five candidates as nodes in a graph, where the node signal (checkmark or cross) is the model's decision for the corresponding candidate, and the edge weights are indicated by the thickness of the connecting line. Alice and Charlie have the same qualifications and are therefore connected with a large edge-weight. For the predictions to satisfy IF, the graph needs to be smooth, i.e., the similar / connected candidates should have similar node signals, which can be accomplished by also offering the job to Alice. In contrast, directly enforcing IF constraints [6] requires a certain degree of output similarity on *all* pairs of candidates, and not just on those which are connected and thus similar. Our main contributions are summarized below.

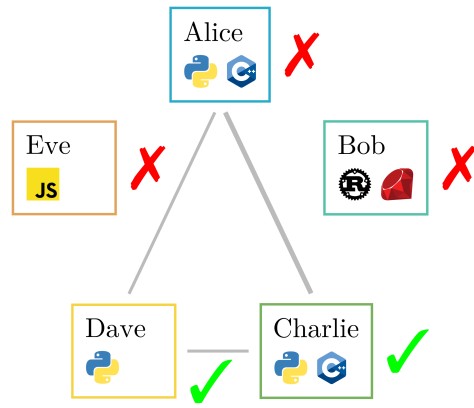

Figure 1: IF on a graph.

1. We cast post-processing for individual fairness as a graph smoothing problem and propose a coordinate descent algorithm to scale the approach to large data sets.
2. We demonstrate theoretically and verify empirically that graph smoothing enforces individual fairness constraints *locally*, i.e., it guarantees similar treatment of *similar* individuals.
3. We empirically compare the Laplacian smoothing method to the post-processing adaptation of the algorithm by Dwork *et al.* [6] enforcing global Lipschitz continuity. The Laplacian smoothing method is not only computationally more efficient but is also more effective in reducing algorithmic bias while preserving accuracy of the original model.
4. We demonstrate the efficacy of Laplacian smoothing on two large-scale text data sets by reducing biases in fine-tuned BERT models.

## 2  Post-processing Problem Formulation

Let $\mathcal{X}$ be the feature space, $\mathcal{Y}$ be the set of possible labels/targets, and $h : \mathcal{X} \to \mathcal{Y}$ be a (possibly unfair) ML model trained for the task. Our goal is to post-process the outputs of $h$ so that they are individually fair. Formally, the post-processor is provided with a set of inputs $\{x_i\}_{i=1}^n$ and the outputs of $h$ on the inputs $\{\widehat{y}_i \triangleq h(x_i)\}_{i=1}^n$, and its goal is to produce $\{\widehat{f}_i\}_{i=1}^n$ that is both individually fair and similar to the $\widehat{y}_i$'s. Recall that individual fairness of $h$ is the Lipschitz continuity of $h$ with respect

to a fair metric $d_{\mathcal{X}}$ on the input space:

$$d_{\mathcal{Y}}(h(x), h(x')) \leq L\, d_{\mathcal{X}}(x, x') \ \text{ for all } x, x' \in \mathcal{X}, \tag{2.1}$$

where $L > 0$ is a Lipschitz constant. The fair metric encodes problem-specific intuition of which samples should be treated similarly by the ML model. It is analogous to the knowledge of protected attributes in group fairness needed to define corresponding fairness constraints. Recent literature proposes several practical methods for learning fair metric from data [18], [19]. We assume the post-processor is either given access to the fair metric (it can evaluate the fair distance on any pair of points in $\mathcal{X}$), or receives feedback on which inputs should be treated similarly. We encode this information in an adjacency matrix $W \in \mathbb{R}^{n \times n}$ of a graph with individuals as nodes. If the post-processor is given the fair metric, then the entries of $W$ are

$$W_{ij} = \begin{cases} \mathsf{exp}(-\theta d_{\mathcal{X}}(x_i, x_j)^2) & d_{\mathcal{X}}(x_i, x_j) \leq \tau \\ 0 & \text{otherwise}, \end{cases} \tag{2.2}$$

where $\theta > 0$ is a scale parameter and $\tau > 0$ is a threshold parameter. If the post-processor is given an annotator's feedback, then $W$ is a binary matrix with $W_{ij} = 1$ if $i$ and $j$ are considered to be treated similarly by the annotator and 0 otherwise. Extensions to multiple annotators are straightforward.

We start with a simple post-processing adaptation of the algorithm by Dwork *et al.* [6] for enforcing individual fairness, that projects the (possibly unfair) outputs of $h$ onto a constraint set to enforce (2.1). In other words, the post-processor seeks the closest set of outputs to the $\widehat{y}_i$'s that satisfies individual fairness:

$$\{\widehat{f}_i\}_{i=1}^n \in \begin{cases} \arg\min_{f_1, \ldots, f_n} & \sum_{i=1}^n \frac{1}{2} d_{\mathcal{Y}}(f_i, \widehat{y}_i)^2 \\ \text{subject to} & d_{\mathcal{Y}}(f_i, f_j) \leq L d_{\mathcal{X}}(x_i, x_j) \end{cases}. \tag{2.3}$$

This objective function, though convex, scales poorly due to the order of $n^2$ constraints. Empirically, we observe that (2.3) leads to post-processed outputs that are dissimilar to the $\widehat{y}_i$'s, leading to poor performance in practice. The goal of our method is to improve performance and scalability, while preserving the IF desiderata of treating *similar* individual similarly. Before presenting our method, we discuss other post-processing perspectives that differ in their applicability and input requirements.

## 2.1 Alternative Post-processing Formulations

We review three post-processing problem setups and the corresponding methods in the literature. First, one can fine-tune a model via an in-processing algorithm to reduce algorithmic biases. Yurochkin *et al.* [12] proposed an in-processing algorithm for IF and used it to train fair models for text classification using sentence BERT embeddings. This setting is the most demanding in terms of input and computational requirements: a user needs access to the original model parameters, fair metric function, and train a predictor, e.g., a moderately deep fully connected neural network, with a non-trivial fairness-promoting objective function.

Second, it is possible to post-process by training additional models to correct the initial model's behavior. For example, Kim *et al.* [20] propose a boosting-based method for group fairness post-processing. This perspective can be adapted to individual fairness; however, it implicitly assumes that we can train weak-learners to boost. Lohia *et al.* [21], [22] propose to train a bias detector to post-process for group fairness and a special, group based, notion of individual fairness. Such methods are challenging to apply to text data or other non-tabular data types.

The third perspective is the most generic: a user has access to original model outputs only, and a minimal additional feedback guiding fairness constraints. Wei *et al.* [23] consider such setting and propose a method to satisfy group fairness constraints; however, it is not applicable to individual fairness. Our problem formulation belongs to this post-processing setup. The main benefit of this approach is its broad applicability and ease of deployment.

## 3 Graph Laplacian Individual Fairness

To formulate our method, we cast IF post-processing as a graph smoothing problem. Using the fair metric or human annotations as discussed in Section 2, we obtain an $n \times n$ matrix $W$ that we treat as an adjacency matrix. As elaborated earlier, the goal of post-processing is to obtain a model $f$ that

is individually fair and accurate. The accuracy is achieved by minimizing the distance between the outputs of $f$ and $h$, a pre-trained model assumed to be accurate but possibly biased. Recall that we do not have access to the parameters of $h$, but can evaluate its predictions. Our method enforces fairness using a graph Laplacian quadratic form [24] regularizer:

$$\widehat{\mathbf{f}} = \arg\min_{\mathbf{f}} \ g_\lambda(\mathbf{f}) = \arg\min_{\mathbf{f}} \ \|\mathbf{f} - \hat{\mathbf{y}}\|_2^2 + \lambda \, \mathbf{f}^\top \mathbb{L}_n \mathbf{f}, \tag{3.1}$$

where $\hat{\mathbf{y}}$ is the output of the model $h$, and $\widehat{\mathbf{f}}$ is the vector of the post-processed outputs, i.e., $\widehat{f}_i = f(x_i)$ for $i = 1, \ldots, n$. The matrix $\mathbb{L}_n \in \mathbb{R}^{n \times n}$ is called *graph Laplacian* matrix and is a function of $W$. There are multiple versions of $\mathbb{L}_n$ popularized in the graph literature (see, e.g., [25] or [26]). To elucidate the connection to individual fairness, consider the unnormalized Laplacian $\mathbb{L}_{un,n} = D - W$, where $D_{ii} = \sum_{j=1}^n W_{ij}$, $D_{ij} = 0$ for $i \neq j$ is the *degree matrix* corresponding to $W$. Then a known identity is:

$$\mathbf{f}^\top \mathbb{L}_{un,n} \mathbf{f} = \tfrac{1}{2} \sum_{i \neq j} W_{ij} \left(f_i - f_j\right)^2. \tag{3.2}$$

Hence, the Laplacian regularizer is small if the post-processed model outputs $\widehat{f}_i$ and $\widehat{f}_j$ (i.e., treatment) are similar for large $W_{ij}$ (i.e., for similar individuals $i$ and $j$). This promotes the philosophy of individual fairness: "treat similar individuals similarly". This observation intuitively explains the motivation for minimizing the graph Laplacian quadratic form to achieve IF. In Section 4, we present a more formal discussion on the connections between the graph Laplacian regularization and IF.

Our post-processing problem (3.1) is easy to solve: setting the gradient of $g_\lambda$ to 0 implies that the optimal solution $\widehat{\mathbf{f}}$ is:

$$\widehat{\mathbf{f}} = \left(I + \lambda \left(\frac{\mathbb{L}_n + \mathbb{L}_n^\top}{2}\right)\right)^{-1} \widehat{\mathbf{y}}. \tag{3.3}$$

The Laplacian $\mathbb{L}_n$ is a positive semi-definite matrix ensuring that (3.1) is strongly convex and that (3.3) is a global minimum. In comparison to the computationally expensive constraint optimization problem (2.3), this approach has a simple closed-form expression.

Note that the symmetry of the unnormalized Laplacian $\mathbb{L}_{un,n}$ simplifies (3.3); however, there are also non-symmetric Laplacian variations. In this work, we also consider the normalized random walk Laplacian $\mathbb{L}_{nrw,n} = (I - \widetilde{D}^{-1}\widetilde{W})$, where $\widetilde{W} = D^{-1/2} W D^{-1/2}$ is the normalized adjacency matrix and $\widetilde{D}$ is its degree matrix. We discuss its properties in the context of IF in Section 4. Henceforth, we refer to our method as Graph Laplacian Individual Fairness (GLIF) when using the unnormalized Laplacian, and GLIF-NRW when using Normalized Random Walk Laplacian.

## 3.1 Prior Work on Graph Laplacians

Graph-based learning via a similarity matrix is prevalent in statistics and ML literature, specifically, in semi-supervised learning. The core idea is to gather information from similar unlabeled inputs to improve prediction accuracy (e.g., see [27], [28], [29] and references therein). Laplacian regularization is widely used in science engineering. We refer to Chapelle [17] for a survey.

We note that [30], [31] also use graph Laplacian regularizers to enforce individual fairness. Our work builds on their work by elucidating the key role played by the graph Laplacian in enforcing individual fairness. In particular, we clarify the connection between the choice of the graph Laplacian and the exact notion of individual fairness the corresponding graph Laplacian regularizer enforces.

## 3.2 Extensions of the Basic Method

In this subsection, we present four extensions of our method: multi-dimensional outputs, coordinate descent for large-scale data, an inductive setting, and alternative output space discrepancy measures.

### 3.2.1 Multi-dimensional Output

We presented our objective function (3.1) and post-processing procedure (3.3) for the case of univariate outputs. This covers regression and binary classification. Our method readily extends to multi-dimensional output space, for example, in classification, $f_i, \widehat{y}_i \in \mathbb{R}^K$ can represent logits, i.e.,

softmax inputs, of the $K$ classes. In this case, $\mathbf{f}$ and $\hat{\mathbf{y}}$ are $n \times K$ matrices, and the term $\mathbf{f}^\top \mathbb{L}_n \mathbf{f}$ is a $K \times K$ matrix. We use the trace of it as a regularizer. The optimization problem (3.1) then becomes:

$$\widehat{\mathbf{f}} = \arg\min_f \; g_\lambda(\mathbf{f}) = \arg\min_f \; \|\mathbf{f} - \hat{\mathbf{y}}\|_F^2 + \lambda \, \mathsf{tr}\left(\mathbf{f}^\top \mathbb{L}_n \mathbf{f}\right), \tag{3.4}$$

where $\|\cdot\|_F$ is the Frobenius norm. Similar to the univariate output case, this yields:

$$\widehat{\mathbf{f}} = \left(I + \lambda\left(\frac{\mathbb{L}_n + \mathbb{L}_n^\top}{2}\right)\right)^{-1} \hat{\mathbf{y}}. \tag{3.5}$$

The solution is the same as (3.3); however, now it accounts for multi-dimensional outputs.

### 3.2.2 Coordinate Descent for Large-Scale Data

Although our method has a closed form solution, it is not immediately scalable, as we have to invert a $n \times n$ matrix to obtain the optimal solution. We propose a *coordinate descent* variant of our method that readily scales to any data size. The idea stems primarily from the gradient of equation (3.4), where we solve:

$$\mathbf{f} - \widehat{\mathbf{y}} + \lambda \frac{\mathbb{L}_n + \mathbb{L}_n^\top}{2}\mathbf{f} = 0. \tag{3.6}$$

Fixing $\{f_j\}_{j\neq i}$, we can solve (3.6) for $f_i$:

$$f_i \leftarrow \frac{\hat{y}_i - \frac{\lambda}{2}\sum_{j\neq i}(\mathbb{L}_{n,ij} + \mathbb{L}_{n,ji})f_j}{1 + \lambda\mathbb{L}_{n,ii}}. \tag{3.7}$$

This gives rise to the coordinate descent algorithm. We perform asynchronous updates over randomly selected coordinate batches until convergence. We refer the reader to Wright [32] and the references therein for the convergence properties of (asynchronous) coordinate descent.

### 3.2.3 Extension to the Inductive Setting

This coordinate descent update is key to extending our approach to the inductive setting. To handle new unseen points, we assume we have a set of test points on which we have already post-processed the outputs of the ML model. To post-process new unseen points, we simply fix the outputs of the other test points and perform a single coordinate descent step with respect to the output of the new point. Similar strategies are often employed to extend transductive graph-based algorithms to the inductive setting [17].

### 3.2.4 Alternative Discrepancy Measures on the Output Space

So far, we have considered the squared Euclidean distance as a measure of discrepancy between outputs. This is a natural choice for post-processing models with continuous-valued outputs. For models that output a probability distribution over the possible classes, we consider alternative discrepancy measures on the output space. It is possible to replace the squared Euclidean distance with a Bregman divergence with very little change to the algorithm in the case of the unnormalized Laplacian. Below, we work through the details for the KL divergence as a demonstration of the idea. A result for the general Bregman divergence can be found in Appendix B.3 (see Theorem B.4).

Suppose the output of the pre-trained model $h$ is $\hat{y}_i \in \Delta^K$, where $\hat{y}_i = \{e^{o_{i,j}}/\sum_{k=1}^K e^{o_{i,k}}\}_{j=1}^K$ a $K$-dimensional probability vector corresponding to a $K$ class classification problem ($\{o_{i,j}\}$ is the output of the penultimate layer of the pre-trained model and $\hat{y}_i$ is obtained by passing it through softmax) and $\Delta^K = \{x \in \mathbb{R}^K : x_i \geq 0, \sum_{i=1}^K x_i = 1\}$ is the probability simplex in $\mathbb{R}^K$. Let $P_v$ denote the multinomial distribution with success probabilities $v$ for any $v \in \Delta^k$. Define $\hat{\eta}_i \in \mathbb{R}^{K-1}$ (resp. $\eta_i$) as the natural parameter corresponding to $\hat{y}_i$ (resp. $f_i$), i.e., $\hat{\eta}_{i,j} = \log(\hat{y}_{i,j}/\hat{y}_{i,K}) = o_{i,j} - o_{i,K}$ for $1 \leq j \leq K-1$. The (unnormalized) Laplacian smoothing problem with the KL divergence is

$$(\tilde{y}_1, \ldots, \tilde{y}_n) = \arg\min_{y_1,\ldots,y_n \in \Delta^K}\left[\sum_i \left\{\mathsf{KL}\left(P_{y_i}||P_{\hat{y}_i}\right) + \frac{\lambda}{2}\sum_{j=1,j\neq i}^n W_{ij}\mathsf{KL}\left(P_{y_i}||P_{y_j}\right)\right\}\right]. \tag{3.8}$$

A coordinate descent approach for solving the above equation is:

$$\tilde{y}_i = \arg\min_{y \in \Delta^k}\left\{\mathsf{KL}\left(P_y||P_{\hat{y}_i}\right) + \frac{\lambda}{2}\sum_{j=1,j\neq i}^n W_{ij}\mathsf{KL}\left(P_y||P_{\tilde{y}_j}\right)\right\}. \tag{3.9}$$

The following theorem establishes that (3.5) solves the above problem in the logit space, or equivalently in the space of the corresponding natural parameters (see Appendix B for the proof):

**Theorem 3.1.** *Consider the following optimization problem on the space of natural parameters:*

$$\tilde{\eta}_i = \arg\min_\eta \left[ \|\eta - \hat{\eta}_i\|^2 + \tfrac{\lambda}{2} \sum_{j=1, j\neq i}^n W_{ij} \|\eta - \tilde{\eta}_j\|^2 \right]. \tag{3.10}$$

*Then, the minimizer $\tilde{\eta}_i$ of equation* (3.10) *is the natural parameter corresponding to the minimizer $\tilde{y}_i$ of* (3.8).

# 4 Local IF and Graph Laplacian Regularization

In this section, we provide theoretical insights into why the graph Laplacian regularizer enforces individual fairness. As pointed out in Section 2, enforcing IF globally is expensive and often reduces a significant amount of accuracy of the final classifier. Here, we establish that solving (3.1) is tantamount to enforcing a localized version of individual fairness, namely *Local Individual Fairness*, which is defined below:

**Definition 4.1** (Local Individual Fairness)**.** *An ML model $h$ is said to be* locally individually fair *if it satisfies:*

$$\mathbb{E}_{x\sim P} \left[ \limsup_{x': d_\mathcal{X}(x,x')\downarrow 0} \frac{d_\mathcal{Y}(h(x), h(x'))}{d_\mathcal{X}(x,x')} \right] \leq L < \infty. \tag{4.1}$$

For practical purposes, this means that $h$ is locally individually fair with constants $\epsilon$ and $L$ if it satisfies

$$d_\mathcal{Y}(h(x), h(x')) \leq L\, d_\mathcal{X}(x, x') \ \text{ for all } x, x' \in \mathcal{X} \text{ where } d_\mathcal{X}(x, x') \leq \epsilon \tag{4.2}$$

in analogy to equation (2.1). Equation (4.2) is a relaxation of traditional IF, where we only care about the Lipschitz-constraint for all pairs of points with small fair distances, i.e., where it is less than some user-defined threshold $\epsilon$.

**Example 4.2.** *For our theoretical analysis, we need to specify a functional form of the fair metric. A popular choice is a Mahalanobis fair metric proposed by [19], which is defined as:*

$$d_\mathcal{X}^2(x, x') = (x - x')^\top \Sigma (x - x'), \tag{4.3}$$

*where $\Sigma$ is a dispersion matrix that puts lower weight in the directions of sensitive attributes and higher weight in the directions of relevant attributes. [19] also proposed several algorithms to learn such a fair metric from the data. If we further assume $d_\mathcal{Y}(y_1, y_2) = |y_1 - y_2|$, then a simple application of Lagrange's mean value theorem yields:*

$$\limsup_{x': d_\mathcal{X}(x,x')\downarrow 0} \frac{|h(x) - h(x')|}{d_\mathcal{X}(x, x')} \leq \|\Sigma^{-1/2} \nabla h(x)\|. \tag{4.4}$$

*This immediately implies:*

$$\mathbb{E}_{x\sim P} \left[ \limsup_{x': d_\mathcal{X}(x,x')\downarrow 0} \frac{d_\mathcal{Y}(h(x), h(x'))}{d_\mathcal{X}(x, x')} \right] \leq \mathbb{E}[\|\Sigma^{-1/2} \nabla h(x)\|], \tag{4.5}$$

*i.e., $h$ satisfies* local individual fairness *constraint as long as $\mathbb{E}[\|\Sigma^{-1/2} \nabla h(x)\|] < \infty$. On the other hand, the global IF constraint necessitates $\sup_{x\in\mathcal{X}} \|\Sigma^{-1/2} \nabla h(x)\| < \infty$, i.e., $h$ is Lipschitz continuous with respect to the Mahalanobis distance.*

The main advantage of this local notion of IF over its global counterpart is that the local definition concentrates on the input pairs with smaller fair distance and ignores those with larger distance. For example, in Figure 1, the edge-weights among Alice, Charlie, and Dave are much larger than among any other pairs (which have a weight of $0$); therefore, our local notion enforces fairness constraint on the corresponding similar pairs, while ignoring (or being less stringent on) others. This prevents over-smoothing and consequently preserves accuracy while enforcing fairness as is evident from our real data experiments in Section 5.

We now present our main theorem, which establishes that, under certain assumptions on the underlying hypothesis class and the distribution of inputs, the graph Laplacian regularizers (both unnormalized

and normalized random walk) enforce the local IF constraint (as defined in Definition 4.1) in the limit. For our theory, we work with $d_{\mathcal{X}}$ as the Mahalanobis distance introduced in Example 4.2 in equation (2.2) along with $\theta = 1/(2\sigma^2)$ ($\sigma$ is a bandwidth parameter which goes to 0 at an appropriate rate as $n \to \infty$) and $\tau = \infty$. All our results will be thorough for any finite $\tau$ but with more tedious technical analysis. Therefore, our weight matrix $W$ becomes:

$$W_{ij} = \frac{|\Sigma|^{1/2}}{(2\pi)^{d/2}\sigma^d}\exp\left(-\frac{1}{2\sigma^2}(x_i - x_j)^\top \Sigma (x_i - x_j)\right). \tag{4.6}$$

The constant $|\Sigma|^{1/2}/((2\pi)^{d/2}\sigma^d)$ is for the normalization purpose and can be absorbed into the penalty parameter $\lambda$. We start by listing our assumptions:

**Assumption 4.3** (Assumption on the domain). *The domain of the inputs $\mathcal{X}$ is a compact subset of $\mathbb{R}^d$ where $d$ is the underlying dimension.*

**Assumption 4.4** (Assumption on the hypothesis). *All functions $f \in \mathcal{F}$ of the hypothesis class satisfy the following:*

*1. The $i^{th}$ derivative $f^{(i)}$ is uniformly bounded over the domain $\mathcal{X}$ of inputs for $i \in \{0, 1, 2\}$.*
*2. $f^{(1)}(x) = 0$ for all $x \in \partial X$, where $\partial X$ denotes the boundary of $\mathcal{X}$.*

**Assumption 4.5** (Assumption on the density of inputs). *The density $p$ of the input random variable $x$ on the domain $\mathcal{X}$ satisfies the following:*

*1. There exists $p_{\max} < \infty$ and $p_{\min} > 0$ such that, for all $x \in \mathcal{X}$, we have $p_{\min} \leq p(x) \leq p_{\max}$.*
*2. The derivatives $\{p^{(i)}\}_{i=0,1,2}$ of the density $p$ are uniformly bounded on the domain $\mathcal{X}$.*

**Discussion on the assumptions** Most of our assumptions (e.g., compactness of the domain, bounded derivatives of $f$ or $p$) are for technical simplicity and are fairly common for the asymptotic analysis of graph regularization (see, e.g., Hein *et al.* [25], [33] and references therein). It is possible to relax some of the assumptions: for example, if the domain $\mathcal{X}$ of inputs is unbounded, then the target function $f$ and the density $p$ should decay at certain rate so that observations far away will not be able to affect the convergence (e.g., sub-exponential tails). Part (2.) of Assumption 4.4 can be relaxed if we assume $p(x)$ is 0 at boundary. However, we do not pursue these extensions further in this manuscript, as they are purely technical and do not add anything of significance to the main intuition of the result.

**Theorem 4.6.** *Under Assumptions 4.3 - 4.5, we have:*

*1. If the sequence of bandwidths $\sigma \equiv \sigma_n \downarrow 0$ such that $n\sigma_n^2 \to \infty$ and $\mathbb{L}_{un,n}$ is unnormalized Laplacian matrix, then*

$$\frac{2}{n^2\sigma^2}\mathbf{f}^\top \mathbb{L}_{un,n}\mathbf{f} \xrightarrow{P} \mathbb{E}_{x\sim p}\left[\nabla f(x)^\top \Sigma^{-1}\nabla f(x)\, p(x)\right]. \tag{4.7}$$

*2. If the sequence of bandwidths $\sigma \equiv \sigma_n \downarrow 0$ such that $(n\sigma^{d+4})/(\log(1/\sigma)) \to \infty$ and $\mathbb{L}_{nrw,n}$ is the normalized random walk Laplacian matrix, then:*

$$\frac{1}{n\sigma^2}\mathbf{f}^\top \mathbb{L}_{nrw,n}\mathbf{f} \xrightarrow{P} \mathbb{E}_{x\sim p}\left[\nabla f(x)^\top \Sigma^{-1}\nabla f(x)\right]. \tag{4.8}$$

*where $\mathbf{f} = \{f(x_i)\}_{i=1}^n$. Consequently, both Laplacian regularizers asymptotically enforce local IF.*

The proof of the above theorem can be found in Appendix B. When we use a normalized random walk graph Laplacian matrix $\mathbb{L}_{nrw,n}$ as regularizer, the regularizer does (asymptotically) penalize $\mathbb{E}\left[\nabla f(x)^\top \Sigma^{-1}\nabla f(x)\right] = \mathbb{E}\left[\|\Sigma^{-1/2}\nabla f(x)\|^2\right]$, which, by Example 4.2, is equivalent to enforcing the local IF constraint. Similarly, the un-normalized Laplacian matrix $\mathbb{L}_{un,n}$, also enforces the same under Assumption 4.5 as:

$$\mathbb{E}\left[\|\Sigma^{-1/2}\nabla f(x)\|^2\right] \leq \frac{1}{p_{\min}}\mathbb{E}\left[\nabla f(x)^\top \Sigma^{-1}\nabla f(x)\, p(x)\right], \text{ where } p_{\min} = \inf_{x\in\mathcal{X}} p(x). \tag{4.9}$$

Although both the Laplacian matrices enforce local IF, the primary difference between them is that the limit of the unnormalized Laplacian involves the density $p(x)$, i.e., it upweights the high-density region (consequently stringent imposition of fairness constraint), whereas it down-weights the under-represented/low-density region. On the other hand, the limit corresponding to the normalized random walk Laplacian matrix does not depend on $p(x)$ and enforces fairness constraint with equal intensity on the entire input space. We used both regularizers in our experiments, comparing and contrasting their performance on several practical ML problems.

# 5 Experiments

The goals of our experiments are threefold:

1. Exploring the trade-offs between post-processing for local IF with GLIF and post-processing with (global) IF constraints using our adaptation of the algorithm by Dwork *et al.* [6] described in (2.3).
2. Studying practical implications of theoretical differences between GLIF and GLIF-NRW, i.e., different graph Laplacians, presented in Section 4.
3. Evaluating the effectiveness of GLIF in its main application, i.e., computationally light debiasing of large deep learning models such as BERT.

The implementation of this work is available at github.com/Felix-Petersen/fairness-post-processing.

## 5.1 Comparing GLIF and Global IF-constraints

For our first experiment, we consider the sentiment prediction task [34], where our goal is to classify words as having a positive or negative sentiment. The baseline model is a neural network trained with GloVe word embeddings [35]. Following Yurochkin *et al.* [11], we evaluate the model on a set of names and observe that it assigns varying sentiments to names. An individually fair model should assign similar sentiment scores to all names. Further, we observe that there is a gap between average sentiments of names typical for Caucasian and African-American ethnic groups [36], which is violating group fairness. Yurochkin *et al.* [11] propose a fair metric learning procedure for this task using a side data set of names, and an in-processing technique for achieving individual fairness. We use their method to obtain the fair metric and compare post-processing of the baseline model with GLIF, GLIF-NRW and the global IF-constraints method. The test set comprises 663 words from the original task and 94 names. For post-processing, no problem specific knowledge is used. The resulting post-processed predictions for the original test set are used to evaluate accuracy, and the predictions on the names are used for evaluating fairness metrics. Even for this small problem, the global IF-constraints method, i.e., a CVXPY [37] implementation of (2.3), takes 7 minutes to run. Due to the poor scalability of the global IF-constraints method, we can use it only for the study of this smaller data set and can not consider it for the large language model experiments in Section 5.2. For GLIF(-NRW), we implement the closed-form solution (3.3) that takes less than a tenth of a second to run. See Appendix A for additional experimental details and a runtime analysis.

We evaluate the fairness-accuracy trade-offs for a range of threshold parameters $\tau$ (for GLIF and GLIF-NRW) and for a range of Lipschitz-constants $L$ (for IF-constraints) in Figure 2. Figure 2 (left) shows the standard deviation of the post-processed outputs on all names as a function of test accuracy on the original sentiment task. Lower standard deviations imply that all names received similar predictions, which is the goal of individual fairness. Figure 2 (center) visualizes group fairness and accuracy, i.e., difference in average name sentiment scores for the two ethnic groups. In this problem, individual fairness is a stronger notion of fairness: achieving similar predictions for all names implies similar group averages, but not vice a versa. Therefore, for this task, post-processing for individual fairness also corrects group disparities.

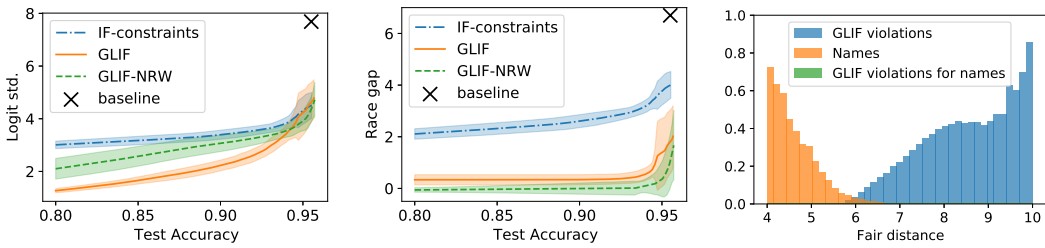

Figure 2: Sentiment experiment. Left: Trade-off between standard deviations of logits of names (measuring individual fairness) and accuracy. Center: Trade-off between race gap (measuring group fairness) and accuracy. Right: Frequencies of violations of the global IF constraints after applying GLIF, constraints corresponding to names, and GLIF's global IF constraint violations for names.

In both settings, GLIF and GLIF-NRW achieve substantially better fairness metrics for the same levels of test accuracy in comparison to the IF-constraints method. To understand the reason for this, we study which global IF constraints are violated after applying the GLIF method in Figure 2 (right). Corresponding to the unique pairs of words in our test set, there are $n(n-1)/2$ unique constraints in (2.3), and the global IF-constraints method satisfies all of them by design. Each constraint (i.e., each pair of words) corresponds to a fair distance, which is small for (under the fair metric) similar words and large for dissimilar words. We bin the constraints by fair distance and present the proportion of global IF constraints violated after applying the GLIF method for each bin in the histogram in Figure 2 (right). Here, we set the Lipschitz-constant $L$ in (2.3) to $L = 2.25$ corresponding to a $89.4\%$ accuracy of the IF-constraints method and show global IF constraint violations of GLIF corresponding to $95\%$ accuracy in blue. This means that we use strong global IF constraints and use a setting of the GLIF method which maintains most of the accuracy, which would not be possible using the IF-constraints method. GLIF does not violate any constraints corresponding to small fair distances, i.e., it satisfies IF on similar individuals, while violating many large fair distance constraints. This can be seen as basically all constraint violations (blue) are at large fair distances of greater or equal 6. This demonstrates the effect of enforcing *local* individual fairness from our theoretical analysis in Section 4. At the same time, we display frequency of constraints that correspond to pairs of names in orange, where we can see that almost all constraints corresponding to names occur at small fair distances of smaller or equal to 6. This is expected in this task because we consider all names similar, so fair distances between them should be small. We can see that the distributions of constraint violations after applying GLIF (blue, right) and names (orange, left) are almost disjoint. We mark all global IF constraint violations after applying GLIF that correspond to names in green, and observe that there are none. Summarizing, GLIF ignores unnecessary (in the context of this problem) constraints allowing it to achieve higher accuracy, while satisfying the more relevant local IF constraints leading to improved fairness.

Regarding the practical differences between GLIF and GLIF-NRW, in Figure 2 (left) GLIF has smaller standard deviations on the name outputs, but in in Figure 2 (center) GLIF-NRW achieves lower race gap. In Theorem 4.6, we showed that GLIF penalizes fairness violations in high density data regions stronger. As a result, GLIF may favor enforcing similar outputs in the high density region causing lower standard deviation, while leaving outputs nearly unchanged in the lower density region, resulting in larger race gaps. GLIF-NRW weights all data density regions equally, i.e., it is less likely to miss a small subset of names, but is less stringent in the high density regions.

### 5.2 Post-processing for Debiasing Large Language Models

Large language models have achieved impressive results on many tasks; however, there is also significant evidence demonstrating that they are prone to biases [4], [38], [39]. Debiasing these models remains largely an open problem: most in-processing algorithms are not applicable or computationally prohibitive due to large and highly complex model architectures, and challenges in handling text inputs. Even if an appropriate in-processing algorithm arises, significant environmental impact due to re-training is unavoidable [4], [16]. In our experiments, we evaluate effectiveness of GLIF as a simple post-processing technique to debias BERT-based models for text classification. Another possible solution is to fine-tune BERT with an in-processing technique as was done by Yurochkin *et al.* [12]. The two approaches are not directly comparable: fine-tuning with SenSeI [12] requires knowledge of the model parameters, alleviates only part of the computational burden, and

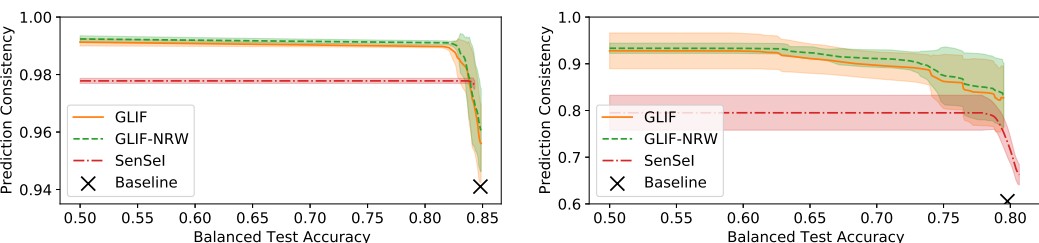

Figure 3: Accuracy-Consistency trade-offs for Bios (left) and Toxicity (right).

| Table 1: Results for the Bios task. | | |
| --- | --- | --- |
| Method | Test Acc. | Pred. Consist. |
| Baseline | **0.846** ± 0.003 | 0.942 ± 0.002 |
| GLIF | 0.830 ± 0.004 | 0.986 ± 0.002 |
| GLIF-NRW | 0.834 ± 0.003 | **0.988** ± 0.002 |
| SenSEI | 0.843 ± 0.003 | 0.977 ± 0.001 |

| Table 2: Results for the Toxicity task. | | |
| --- | --- | --- |
| Method | Test Acc. | Pred. Consist. |
| Baseline | **0.809** ± 0.004 | 0.614 ± 0.013 |
| GLIF | 0.803 ± 0.003 | 0.835 ± 0.012 |
| GLIF-NRW | 0.803 ± 0.003 | **0.844** ± 0.013 |
| SenSEI | 0.791 ± 0.005 | 0.773 ± 0.043 |

has more stringent requirements on the fair metric, while post-processing with GLIF is transductive, i.e., it requires access to unlabeled test data (see extended discussion in Section 2.1).

We replicate the experiments of Yurochkin *et al.* [12] on Bios [40] and Toxicity[1] data sets. They use the approach of Mukherjee *et al.* [19] for fair metric learning which we reproduce. We refer to the Appendix B.1 of [12] for details. In both tasks, following [12], we quantify performance with balanced accuracy due to class imbalance, and measure individual fairness via *prediction consistency*, i.e., the fraction of test points where the prediction remains unchanged when performing task-specific input modifications. For implementation details, see Appendix A. In Appendix A.4, we analyze the runtime and distinguish between the closed-form and coordinate descent variants of GLIF.

In *Bios*, the goal is to predict the occupation of a person based on their textual biography. Such models can be useful for recruiting purposes. However, due to historical gender bias in some occupations, the baseline BERT model learns to associate gender pronouns and names with the corresponding occupations. Individual fairness is measured with prediction consistency with respect to gender pronouns and names alterations. A prediction is considered consistent if it is the same after swapping the gender pronouns and names. We present the fairness-accuracy trade-off in Figure 3 (left) for a range of threshold parameters $\tau$, and compare performance based on hyperparameter values selected with a validation data in Table 1. Both GLIF and GLIF-NRW noticeably improve individual fairness measured with prediction consistency, while retaining most of the accuracy.

In *Toxicity*, the task is to identify toxic comments—an important tool for facilitating inclusive discussions online. The baseline BERT model learns to associate certain identity words with toxicity (e.g., "gay") because they are often abused in online conversations. The prediction consistency is measured with respect to changes to identity words in the inputs. There are 50 identity words, e.g., "gay", "muslim", "asian", etc. and a prediction is considered consistent if it is the same for all 50 identities. We present the trade-off plots in Figure 3 (right) and compare performance in Table 2 (right). Our methods reduce individual biases in BERT predictions. We note that in both Toxicity and Bios experiments, we observe no practical differences between GLIF and GLIF-NRW.

## 6   Summary and Discussion

We studied post-processing methods for enforcing individual fairness. The methods provably enforce a local form of IF and scale readily to large data sets. We hope this broadens the appeal of IF by (i) alleviating the computational costs of operationalizing IF and (ii) allowing practitioners to use off-the-shelf models for standard ML tasks. We also note that it is possible to use our objective for in-processing.

We conclude with two warnings: First, enforcing any algorithmic fairness definition does not guarantee complete fairness from the perspective of the user. The problem-specific meaning of fairness is often hard to encode exactly with a mathematical fairness definition. Second, while local individual fairness is a reasonable choice in many applications, this choice should be understood and verified by the practitioner depending on the situation.

---

[1]Based on the Kaggle "Toxic Comment Classification Challenge".

## Acknowledgments and Disclosure of Funding

This note is based upon work supported by the National Science Foundation (NSF) under grants no. 1916271, 2027737, and 2113373 and supported by the German Research Foundation (DFG) under Germany's Excellence Strategy EXC–2117–390829875. Any opinions, findings, and conclusions or recommendations expressed in this note are those of the authors and do not necessarily reflect the views of the NSF nor the DFG.

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
