# Appendix: Post-processing for Individual Fairness

Felix Petersen*     Debarghya Mukherjee*     Yuekai Sun     Mikhail Yurochkin

In the appendices, we start by explaining experimental details in Appendix A and present the proofs for our main theorems in Appendix B.

## A    Experimental Details

For the experimental evaluation, we use the three methods: IF-constraints, GLIF, and GLIF-NRW (see Section 3). For GLIF and GLIF-NRW, we use both the closed-form solution (see equation (3.3)) and the coordinate descent algorithm (Section 3.2.2). We evaluate on the sentiment, the bios, and the toxicity data sets, which are discussed in the following. For all experiments, we report means and standard deviations over 10 repetitions / seeds.

### A.1    Experimental Settings

**Sentiment prediction**    In this task, we post-process a neural network with $1\,000$ hidden units trained to predict sentiment, e.g., "nice" is positive and "ugly" is negative, using $5\,961$ labeled words embedded with 300-dimensional GloVe word embeddings [35]. Post-processing is applied to the model predictions on 663 (unlabeled) test words mixed with 94 names (49 names popular among Caucasian population and 45 names popular among African-American population). This experiment is based on the fair sentiment prediction experiment of Yurochkin *et al.* [11]. Train and test words were collected by Hu *et al.* [34]. The list of names for evaluating fairness was proposed by Caliskan *et al.* [36]. To obtain the fair metric, we followed the original experiment [11]: learn a "sensitive" subspace via PCA with 50 components applied to a side data set of popular baby names in New York City. The fair metric is constructed to ignore any variation in this subspace, i.e., it is equal to Euclidean distance on word embeddings projected onto the orthogonal complement of the sensitive subspace.

**Bios**    We use the data set proposed by de Arteaga *et al.* [40] and follow the experimental setup of Yurochkin *et al.* [12]. In this task, we post-process fine-tuned BERT-Base-Uncased [14] replicating the training setup described in Appendix B in [12]. This yields 28-dimensional outputs (a logit per class) for each biography. We also replicate fair metric learning procedure of Yurochkin *et al.* [12]: for each training bio, we create an alternative bio by swapping gender pronouns, e.g., "He is a lawyer" to "She is a lawyer", and use the embeddings from the fine-tuned BERT. Then, we use the FACE method by Mukherjee *et al.* [19] with 25 factors, considering each pair of original and altered bios as a pair of similar examples. For each seed, $354\,080$ training biographies are used for fine-tuning BERT and fair metric learning. We apply post-processing to $39\,343$ test predictions mixed with the same number of predictions for bios created by altering names and gender pronouns. Thus, the total number of predictions that we post-process is $78\,686$. We use the altered bios in the test set to evaluate prediction consistency, and evaluate the test accuracy only on the $39\,343$ original test bios. Our methods do not have any knowledge of the alteration procedure nor which bios are the alterations.

**Toxicity**    For this task, we use the data set derived from the "Toxic Comment Classification Challenge" Kaggle competition following the experimental setup of Yurochkin *et al.* [12]. BERT fine-tuning and fair metric learning is similar to the bios experiment and follows the original experiment. This is a binary classification problem. The experiment utilizes a list of 50 identity tokens [41] analogous to the gender pronouns in the Bios experiment. There are $155\,618$ training comments, among

which 55% have at least one of the 50 identity tokens. To learn the fair metric, a random subset of 25 identity tokens (out of the 50) is used. For each comment in the training set (with at least one of the known 25 identity tokens), 25 alterations are created, forming groups of comparable samples for the FACE method with 25 factors. At test time, we are interested in prediction consistency on the original comments and all 50 alterations. For each seed, we apply our methods to post-process predictions on a test set comprising around 3 640 test comments without any of the 50 identity tokens, around 4 550 test comments with at least one of the identity tokens, and their 227 500 alterations corresponding to 50 identity tokens. The total number of post-processed predictions (depending on the seed) is around 235 690. The test accuracy is evaluated on the original (unaltered) $3\,640 + 4\,550 = 8\,190$ test comments. As before, our methods have no knowledge of the identity tokens nor the existence of alterations in the data being post-processed.

## A.2  Methods

For the sentiment task, we use the closed-form method of GLIF as in equation (3.3) of the main text. For the other tasks, we use the coordinate descent algorithm (described in Section 3.2.2 of the main text) as the test data set is too large to fit in memory. For the coordinate descent algorithm, we used 10 epochs, which we found to work well across all data sets.

## A.3  Hyperparameters

We used grid search on a validation data set to find the best hyperparameter for each experimental setting. We optimized the threshold parameter $\tau$ (see equation (2.2)) and the regularization strength $\lambda$ (see equations (3.1) and (3.4)) considering the following ranges:

- $\lambda \in \{0.01, 0.03, 0.1, 0.3, 1, 3, 10, 30, 100\}$
- $\tau \in \{10^{0.02i} \text{ for } i \in \{-50...100\}\}$ and rounded to a close fraction.

For GLIF-NRW, we (internally) multiply $\lambda$ by the average degree in the graph which yields a good effective $\lambda$ for each $\tau$ (as $\tau$ significantly influences the average degree). Using this procedure, we found the following to work best for each data set, which we also used for Tables 1 and 2 in the main text:

**Sentiment** $\lambda = 0.1, \tau = 30$

**Bios** $\lambda = 10, \tau = 16$

**Toxicity** $\lambda = 30, \tau = 0.4$

We found that the exact value is not crucial, and multiple neighboring $\lambda$s and $\tau$s achieved around the same performance. As for the factor $\theta$ (see equation (2.2)), we used $\theta = 10^{-4}$ but found this to perform indistinguishable to other choices such as $10^{-3}$, $10^{-6}$, and $10^{-8}$.

For the accuracy-fairness trade-off plots, we plot the full range of $\tau$.

## A.4  Runtime Analysis

We ran the experiments on a local iMac (3.6 GHz Intel Core i9), single-threaded, and not requiring a GPU. We report runtimes in Table 3. For the sentiment data set, GLIF and GLIF-NRW are four orders of magnitude faster than IF-constraints. IF-constraints is too slow to be practical on the larger data sets, and we did not evaluate it on Bios and Toxicity. To demonstrate the speed trade-off for different number of test points on Bios and Toxicity, we include test sets of 1% and 10% of the original size. We report the runtimes for single-threaded computation. Running it multi-threaded reduces computation time, respectively.

For the closed-form GLIF, the runtime is $\mathcal{O}(n^3)$ where $n$ is the number of test points due to the matrix inversion. Note that the theoretical runtime of matrix inversion is $\mathcal{O}(n^{2.373})$ using optimized Coppersmith-Winograd–like algorithms, but not practical in our settings.

For the coordinate descent GLIF, the runtime is $\mathcal{O}(n^2 \cdot c)$ where $c$ is the number of epochs during coordinate descent. We found that $c = 10$ works well across all data sets. Note that by increasing $n$, the runtime increases because more points have to be updated and (for each point) more potential neighbors have to be considered.

The expected runtimes match the empirical runtimes reported in Table 3.

Table 3: Runtimes on a 3.6 GHz Intel Core i9 iMac.

| Data Set | Sentiment | Bios | | | Toxicity | | |
|---|---|---|---|---|---|---|---|
| # Points | 757 | 788 | 7 870 | 78 686 | 2 383 | 23 620 | 235 842 |
| IF-constraints | 422s | — | — | — | — | — | — |
| Closed Form GLIF | 0.02s | 0.02s | 8.8s | — | 0.33s | 228s | — |
| Closed Form GLIF-NRW | 0.04s | 0.05s | 32s | — | 1.03s | 864s | — |
| Coordinate Desc. GLIF | 0.08s | 0.18s | 30s | 4 569s | 0.94s | 111s | 12 200s |
| Coordinate Desc. GLIF-NRW | 0.09s | 0.19s | 30s | 4 812s | 1.19s | 136s | 13 500s |

# B  Proofs of our Main Theorems

## B.1  Proof of Theorem 4.6

For technical simplicity here we show that if the fair metric is euclidean distance then the un-normalized graph Laplacian regularizer converges to $\mathbb{E}[\|\nabla f(X)\|^2 p(X)]$ and the normalized random walk graph Laplacian regularization converges to $\mathbb{E}[\|\nabla f(X)\|^2]$. As our fair metric (i.e., Mahalanobish distance) is equivalent to euclidean metric in a sense that there exists $c_1, c_2 > 0$ such that:

$$c_1\|x_1 - x_2\| \leq d_{\text{Fair}}(x_1, x_2) = (x_1 - x_2)^\top \Sigma (x_1 - x_2) \leq c_2\|x_1 - x_2\|,$$

where $c_1$ is the minimum eigenvalue of $\Sigma$ and $c_2$ is its maximum eigenvalue, all of our calculations are valid for this fair distance with a tedious tracking of this equivalence. As this proof is itself very involved and this generalization from euclidean to Mahalanobis distance adds nothing of major significance to the core idea of the proof, we confine ourselves to the euclidean distance.

### B.1.1  Proof of Part 1.

*Proof.* For un-normalized graph Laplacian, $\mathbb{L}_{un,n} = D - W$ where:

$$W_{ij} = \frac{1}{(2\pi)^{d/2}h^d}e^{-\frac{1}{2h^2}\|x_i - x_j\|^2}, \quad D_{ii} = \sum_{j=1}^{n} W_{ij}.$$

The regularizer can be reformulated as:

$$\frac{1}{n^2h^2}\mathbf{f}^\top (D - W)\mathbf{f} = \frac{1}{n^2h^2}\sum_{ij}(D - W)_{ij}f(X_i)f(X_j)$$

$$= \frac{1}{n^2h^2}\sum_{i}(D_{ii} - W_{ii})f^2(X_i) - \sum_{i \neq j}W_{ij}f(X_i)f(X_j)$$

$$= \frac{1}{n^2h^2}\sum_{i}\sum_{i \neq j}W_{ij}\left(f^2(X_i) - f(X_i)f(X_j)\right)$$

$$= \frac{1}{2n^2h^2}\sum_{i}\sum_{j \neq i}W_{ij}\left(f(X_i) - f(X_j)\right)^2.$$

Therefore we need to establish:

$$\frac{1}{n(n-1)h^2}\sum_{i}\sum_{j \neq i}W_{ij}\left(f(X_i) - f(X_j)\right)^2 \xrightarrow{P} \mathbb{E}\left[\|\nabla f(X)\|^2 p(X)\right].$$

Towards that direction, we show that the expectation of the random regularizer converges to $\mathbb{E}\left[\|\nabla f(X)\|^2 p(X)\right]$ and its variance goes to 0 under our assumptions. For the expectation:

$$\mathbb{E}\left[\frac{1}{n(n-1)h^2}\sum_{i}\sum_{j \neq i}W_{ij}\left(f(X_i) - f(X_j)\right)^2\right]$$

$$= \frac{1}{h^2} \int_{\mathcal{X}} \int_{\mathcal{X}} \frac{1}{(2\pi)^{d/2}h^d} e^{-\frac{1}{2h^2}\|x-y\|^2} (f(x) - f(y))^2 \, p(y) \, dy \, p(x) \, dx$$

$$= \frac{1}{h^2} \int_{\mathcal{X}} \int_{\frac{\mathcal{X}-x}{h}} \frac{1}{(2\pi)^{d/2}} e^{-\frac{1}{2}\|x-y\|^2} (f(x + hz) - f(x))^2 \, p(x + hz) \, dz \, p(x) \, dx$$

$$= \int_{\mathcal{X}} \int_{\frac{\mathcal{X}-x}{h}} \left[ \frac{1}{(2\pi)^{d/2}} e^{-\frac{1}{2}\|x-y\|^2} \left( z^\top \nabla f(x) + \frac{h}{2} z^\top \nabla^2 f(\widetilde{x}) z \right)^2 \right.$$

$$\left. \times \left( p(x) + h\nabla p(x^*) \right) \, dz \, p(x) \, dx \right] \qquad [\widetilde{x}, x^* \text{ are intermediate points}]$$

$$= \int_{\mathcal{X}} \int_{\frac{\mathcal{X}-x}{h}} \left[ \frac{1}{(2\pi)^{d/2}} e^{-\frac{1}{2}\|x-y\|^2} z^\top \nabla f(x) \nabla f(x)^\top z \, dz \, p^2(x) \, dx \right] + O(h)$$

$$= \int_{\mathcal{X}} \nabla f(x)^\top \mathbb{E}_Z \left[ ZZ^\top \mathbb{1}_{x+Zh \in \mathcal{X}} \right] \nabla f(x) p^2(x) \, dx + O(h) \longrightarrow \mathbb{E} \left[ \|\nabla f(X)\|^2 p(X) \right].$$

where the last line follows from Vitali's theorem as the derivative $\nabla f(x)$ has finite variance. Thus we have proved that the expectation of the regularizer converges to the desired limit. The final step is to show that the variance of the regularizer converges to 0. Towards that direction:

$$\text{var} \left( \frac{1}{n^2 h^2} \sum_i \sum_{j \neq i} W_{ij} (f(X_i) - f(X_j))^2 \right)$$

$$= \frac{1}{n^4 h^4} \sum_{i \neq j} \mathbb{E} \left[ W_{ij}^2 (f(X_i) - f(X_j))^4 \right]$$

$$+ \frac{1}{n^4 h^4} \sum_{(i,j) \neq (k,l)} \text{cov} \left( W_{ij} (f(X_i) - f(X_j))^2, W_{kl} (f(X_k) - f(X_l))^2 \right)$$

$$= O(n^{-2}) + \frac{1}{n^4 h^4} \sum_{(i,j) \neq (k,l)} \text{cov} \left( W_{ij} (f(X_i) - f(X_j))^2, W_{kl} (f(X_k) - f(X_l))^2 \right)$$

$$= O(n^{-2}) + V$$

That the first summand is $O(n^{-2})$ follows from a similar calculation used to establish the convergence of the expectation and hence skipped. For the covariance term $V$, if there is not indices common between $(i,j)$ and $(k,l)$, the covariance term is $0$. Therefore we consider only those terms where there is exactly one index common between $(i,j)$ and $(k,l)$. Therefore:

$$V = \frac{1}{n^4 h^4} \sum_{(i,j) \neq (k,l)} \text{cov} \left( W_{ij} (f(X_i) - f(X_j))^2, W_{kl} (f(X_k) - f(X_l))^2 \right)$$

$$= \frac{1}{n^4 h^4} \sum_{i \neq j \neq k} \text{cov} \left( W_{ij} (f(X_i) - f(X_j))^2, W_{ik} (f(X_i) - f(X_k))^2 \right)$$

$$= \frac{n(n-1)(n-2)}{n^4 h^4} \text{cov} \left( W_{ij} (f(X_i) - f(X_j))^2, W_{ik} (f(X_i) - f(X_k))^2 \right)$$

$$= \frac{n(n-1)(n-2)}{n^4 h^4} \left[ \mathbb{E} \left[ W_{ij} W_{ik} (f(X_i) - f(X_j)) (f(X_i) - f(X_k)) \right] \right.$$

$$\left. - \mathbb{E} \left[ W_{ij} (f(X_i) - f(X_j)) \right] \mathbb{E} \left[ W_{ik} (f(X_i) - f(X_k)) \right] \right]$$

$$= \frac{n(n-1)(n-2)}{n^4 h^4} \mathbb{E} \left[ W_{ij} W_{ik} (f(X_i) - f(X_j)) (f(X_i) - f(X_k)) \right] + O(n^{-1})$$

For the cross term:

$$\mathbb{E} \left[ W_{ij} W_{ik} (f(X_i) - f(X_j)) (f(X_i) - f(X_k)) \right]$$

$$= \int_{\mathcal{X}} \int_{\mathcal{X}} \int_{\mathcal{X}} \left[ \left( \frac{1}{(2\pi)^{d/2}h^d} e^{-\frac{1}{2h^2}\|x-y\|^2} \right) \left( \frac{1}{(2\pi)^{d/2}h^d} e^{-\frac{1}{2h^2}\|x-w\|^2} \right) \right.$$

$$\times (f(x) - f(y))(f(x) - f(w))\, p(w)p(y)p(x)\; dw\, dy\, dx]$$

$$= \int_{\mathcal{X}} \int_{\frac{\mathcal{X}-x}{h}} \int_{\frac{\mathcal{X}-x}{h}} \left[ \left( \frac{1}{(2\pi)^{d/2}} e^{-\frac{1}{2}\|z_1\|^2} \right) \left( \frac{1}{(2\pi)^{d/2}} e^{-\frac{1}{2}\|z_2\|^2} \right) \right.$$

$$\left. \times (f(x) - f(x + hz_1))(f(x) - f(x + hz_2))\, p(x + hz_1)p(x + hz_2)p(x)\; dz_1\, dz_2\, dx \right]$$

$$= h^2 \int_{\mathcal{X}} \int_{\frac{\mathcal{X}-x}{h}} \int_{\frac{\mathcal{X}-x}{h}} \phi(z_1)\phi(z_2) z_1^\top \nabla f(x) \nabla f(x)^\top z_2\, p(z_1)p(z_2)p(x)\; dz_2 dz_1 dx + o(h^2)$$

$$= h^2 \int_{\mathcal{X}} g_h(x)p(x)\; dz_2 dz_1 dx + o(h^2)$$

where the function $g_h(x)$ is defined as:

$$g_h(x) = \nabla f(x)^\top \mathbb{E}_{Z_1, Z_2} \left[ Z_1 Z_2^\top \mathbb{1}_{x + Z_1 h \in \mathcal{X}} \mathbb{1}_{x + Z_2 h \in \mathcal{X}} \right] \nabla f(x).$$

It is immediate that $g_h(x) \to 0$ pointwise for all $x \in \mathcal{X}$. Further, as the derivative of $f$ has finite variance, we also have $g_h$ is uniformly integrable. Therefore, another application of Vitali's theorem yields:

$$\int_{\mathcal{X}} g_h(x)p(x)\; dx \xrightarrow{p} 0 \implies V = o(h^2).$$

This completes the proof. $\qquad\qquad\square$

**Remark B.1.** *The assumption on the bandwidth $h_n$ in the part 1. of Theorem 4.6 can be relaxed upto the condition $nh_n \to \infty$ if we further assume $\nabla f(x) = 0$ or $p(x) = 0$ on the boundary of $\mathcal{X}$.*

### B.1.2 Proof of Part 2.

*Proof.* As in the case of Part 1. here also we confine ourselves to the euclidean distance. Before delving into the technical details, we introduce a few notations for the ease of the proof. The normalized random walk Laplacian regularizer can be expressed as:

$$\frac{1}{nh^2} \mathbf{f}^\top \mathbb{L}_{nrw,n} \mathbf{f} = \frac{1}{nh^2} \mathbf{f}^\top \left( I - \widetilde{D}^{-1} \widetilde{K} \right) \mathbf{f},$$

where:

$$\widetilde{K}_{ij} = \frac{\frac{1}{nh^d} K \left( \frac{\|X_i - X_j\|^2}{h^2} \right)}{\sqrt{\frac{1}{nh^d} \sum_i K \left( \frac{\|X_i - X_j\|^2}{h^2} \right)} \sqrt{\frac{1}{nh^d} \sum_j K \left( \frac{\|X_i - X_j\|^2}{h^2} \right)}}$$

$$:= \frac{\frac{1}{nh^d} K \left( \frac{\|X_i - X_j\|^2}{h^2} \right)}{\sqrt{d_{n,h}(X_i)} \sqrt{d_{n,h}(X_j)}} \qquad \left[ K(z) = \phi(z) = \frac{1}{(2\pi)^{d/2}} e^{-\frac{1}{2}\|z\|^2} \right],$$

$$\widetilde{D}_{ii} = \sum_j \widetilde{K}_{ij} = \frac{1}{nh^d} \sum_j \frac{K \left( \frac{\|X_i - X_j\|^2}{h^2} \right)}{\sqrt{d_{n,h}(X_i)} \sqrt{d_{n,h}(X_j)}} = \frac{1}{\sqrt{d_{n,h}(X_i)}} \frac{1}{nh^d} \sum_j \frac{K \left( \frac{\|X_i - X_j\|^2}{h^2} \right)}{\sqrt{d_{n,h}(X_j)}}$$

We further define few more functions which are imperative for the rest of the proof:

$$d_{n,h}(x) = \frac{1}{nh^d} \sum_i K \left( \frac{\|x - X_i\|^2}{h^2} \right)$$

$$p_h(x) = \mathbb{E}[d_{n,h}(x)] = \mathbb{E} \left[ \frac{1}{h^d} K \left( \frac{\|x - X\|^2}{h^2} \right) \right]$$

$$\widetilde{d}_{n,h}(x) = \frac{1}{n} \sum_{i=1}^n \frac{\frac{1}{h^d} K \left( \frac{\|x - X_i\|^2}{h^2} \right)}{\sqrt{d_{n,h}(x)} \sqrt{d_{n,h}(X_i)}}$$

$$\widetilde{\widetilde{d}}_{n,h}(x) = \frac{1}{n} \sum_{i=1}^n \frac{\frac{1}{h^d} K \left( \frac{\|x - X_i\|^2}{h^2} \right)}{\sqrt{p_h(x)} \sqrt{p_h(X_i)}}$$

$$\widetilde{\widetilde{d}}_h(x) = \mathbb{E}\left[\frac{\frac{1}{h^d}K\left(\frac{\|x-X\|^2}{h^2}\right)}{\sqrt{p_h(x)}\sqrt{p_h(X)}}\right]$$

Following two auxiliary lemmas will be used frequently throughout the proof:

**Lemma B.2.** *The function $p_h(x)$ and $\widetilde{\widetilde{d}}_h(x)$ is uniformly lower bounded over $x \in \mathcal{X}$, i.e., there exists $\widetilde{p}_{\min} > 0$ and $\widetilde{d}_{\min} > 0$ such that $p_h(x) \geq \widetilde{p}_{\min}$ and $\widetilde{\widetilde{d}}_h(x) \geq \widetilde{d}_{\min}$ for all $x \in \mathcal{X}$ uniformly over all small $h$.*

*Proof.* The definition of $p_h(x)$ yields:

$$p_h(x) = \mathbb{E}\left[\frac{1}{h^d}K\left(\frac{\|x-X\|^2}{h^2}\right)\right] = \int_{\mathcal{X}} \frac{1}{h^d}K\left(\frac{\|x-y\|^2}{h^2}\right)p(y)\,dy$$

$$= \int_{\frac{\mathcal{X}-x}{h}} K(\|z\|^2)p(x+hz)\,dz$$

$$\geq p_{\min}\int_{\frac{\mathcal{X}-x}{h}} K(\|z\|^2)\,dz$$

$$\geq p_{\min}\inf_{x\in\mathcal{X}}\int_{\frac{\mathcal{X}-x}{h}} K(\|z\|^2)\,dz := \widetilde{p}_{\min}.$$

Note that, the bound $\widetilde{p}_{\min}$ is independent of $h$ for all small $h$ as the volume of the region $(\mathcal{X}-x)/h$ increases as $h \to 0$. Moreover we can further establish an upper bound on $p_h(x)$:

$$p_h(x) = \int_{\frac{\mathcal{X}-x}{h}} K(\|z\|^2)p(x+hz)\,dz$$

$$\leq p_{\max}\int_{\frac{\mathcal{X}-x}{h}} K(\|z\|^2)\,dz$$

$$\leq p_{\max}\int_{\mathbb{R}^d} K(\|z\|^2)\,dz = p_{\max} \quad \left[\because \int_{\mathbb{R}^d} K(\|z\|^2)\,dz = 1\right].$$

We use the above upper bound on $p_h(x)$ to obtain a lower bound on $\widetilde{\widetilde{d}}_h(x)$ as follows:

$$\widetilde{\widetilde{d}}_h(x) = \mathbb{E}\left[\frac{\frac{1}{h^d}K\left(\frac{\|x-X\|^2}{h^2}\right)}{\sqrt{p_h(x)}\sqrt{p_h(X)}}\right] = \int_{\mathcal{X}} \frac{\frac{1}{h^d}K\left(\frac{\|x-y\|^2}{h^2}\right)}{\sqrt{p_h(x)}\sqrt{p_h(y)}}p(y)\,dy$$

$$\geq \frac{1}{p_{\max}}\int_{\mathcal{X}} \frac{1}{h^d}K\left(\frac{\|x-y\|^2}{h^2}\right)p(y)\,dy$$

$$\geq \frac{\widetilde{p}_{\min}}{p_{\max}} := \widetilde{d}_{\min}.$$

$\square$

**Lemma B.3.** *Under the main assumptions stated in Theorem 4.6, we have:*

$$\sup_{x\in\mathcal{X}}|d_{n,h}(x) - p_h(x)| = O_p\left(\sqrt{\frac{1}{nh^d}\log\frac{1}{h}}\right), \tag{B.1}$$

$$\sup_{x\in\mathcal{X}}\left|\widetilde{d}_{n,h}(x) - \widetilde{\widetilde{d}}_{n,h}(x)\right| = O_p\left(\sqrt{\frac{1}{nh^d}\log\frac{1}{h}}\right), \tag{B.2}$$

$$\sup_{x\in\mathcal{X}}\left|\widetilde{\widetilde{d}}_{n,h}(x) - \widetilde{\widetilde{d}}_h(x)\right| = O_p\left(\sqrt{\frac{1}{nh^d}\log\frac{1}{h}}\right). \tag{B.3}$$

*Therefore combining the bounds of equation* (B.2) *and* (B.3) *we obtain:*

$$\sup_{x\in\mathcal{X}}\left|\widetilde{d}_{n,h}(x) - \widetilde{\widetilde{d}}_h(x)\right| = O_p\left(\sqrt{\frac{1}{nh^d}\log\frac{1}{h}}\right). \tag{B.4}$$

*Proof.* From the definition of $d_{n,h}(x)$ we can write it as $d_{n,h}(x) = \mathbb{P}_n K_{h,x}$ where $K_{h,x}(y) = (1/h^d)K(\|x-y\|^2/h^2)$. This implies $p_h(x) = PK_{h,x}$. Now, for any $x_1, x_2, y \in \mathcal{X}$:

$$\begin{aligned}
|K_{h,x_1}(y) - K_{h,x_2}(y)| &\leq \frac{1}{h^d(2\pi)^{d/2}} \left| K\left(\frac{\|x_1-y\|^2}{h^2}\right) - K\left(\frac{\|x_2-y\|^2}{h^2}\right) \right| \\
&= \frac{1}{h^d(2\pi)^{d/2}} \left| \exp\left(-\frac{\|x_1-y\|^2}{2h^2}\right) - \exp\left(-\frac{\|x_2-y\|^2}{2h^2}\right) \right| \\
&= \frac{1}{h^d(2\pi)^{d/2}} \left| \left\langle x_1 - x_2, -(x^*-y)\frac{1}{h^2}\exp\left(-\frac{\|x^*-y\|^2}{2h^2}\right) \right\rangle \right| \\
&\leq \frac{2}{h^{d+2}(2\pi)^{d/2}} \|x_1-x_2\| \, \|x^*-y\| \exp\left(-\frac{\|x^*-y\|^2}{2h^2}\right) \\
&\leq \frac{2}{h^{d+1}(2\pi)^{d/2}} \|x_1-x_2\| \frac{\|x^*-y\|}{h} \exp\left(-\frac{\|x^*-y\|^2}{2h^2}\right) \\
&\leq \frac{2}{h^{d+1}} \|x_1-x_2\| \sup_z \frac{1}{(2\pi)^{d/2}} \|z\| e^{-\frac{z^2}{2}} \\
&\leq \frac{L}{h^{d+1}} \|x_1-x_2\| \qquad \left[ L = \sup_z \frac{1}{(2\pi)^{d/2}} \|z\| e^{-\frac{z^2}{2}} \right].
\end{aligned}$$

As the above bound is free of $y$, we further have:

$$\|K_{h,x_1} - K_{h,x_2}\|_\infty \leq \frac{L}{h^{d+1}} \|x_1-x_2\|. \tag{B.5}$$

The envelope function of the collection $\mathcal{K} = \{K_{h,x} : x \in \mathcal{X}\}$ is:

$$\bar{K}_h(y) = \sup_x d_x(y) = \sup_x \frac{1}{h^d C} K\left(\frac{\|x-y\|^2}{h^2}\right) = \frac{1}{h^d C} := U.$$

Now fix $\epsilon > 0$. Suppose $\mathcal{X}_{\epsilon,h} := \{x_1, x_2, \ldots, x_N\}$ is $(\epsilon h)/LC$ covering set of $\mathcal{X}$ (which is finite as $\mathcal{X}$ is compact). Then for any $x \in \mathcal{X}$, there exists $x^* \in \mathcal{X}_{\epsilon h}$ such that $\|x - x^*\| \leq \epsilon h$. This along with equation (B.5) implies:

$$\left\| \frac{K_{x,h}}{U} - \frac{K_{x^*,h}}{U} \right\|_\infty \leq \frac{L}{U h^{d+1}} \|x - x^*\| \leq \frac{L}{U h^{d+1}} \frac{\epsilon h}{LC} = \epsilon,$$

i.e.:

$$\sup_Q \mathcal{N}(\mathcal{K}, L_2(Q), \epsilon U) \leq \mathcal{N}(\mathcal{K}, L_\infty, \epsilon U)$$

$$\leq \mathcal{N}\left(\frac{\epsilon h}{LC}, L_2, \mathcal{X}\right)$$

$$\leq K\left(\frac{LC}{\epsilon h}\right)^d := \left(\frac{K_1}{\epsilon h}\right)^d.$$

The maximum variation of functions of $\mathcal{K}$ can be bounded as below:

$$\begin{aligned}
\sup_x \mathsf{var}(K_{h,x}) &\leq \sup_x \mathbb{E}[K_{h,x}^2] \\
&= \sup_x \int \frac{1}{h^{2d}C^2} K^2\left(\frac{\|x-y\|^2}{h^2}\right) p(y)\, dy \\
&\leq \frac{1}{h^d C^2} \sup_x \int K^2\left(\|z\|^2\right) p(x+zh)\, dz \\
&\leq \frac{p_{\max}}{h^d C^2} \int K^2\left(\|z\|^2\right)\, dz := \frac{K_2}{h^d} := \sigma^2.
\end{aligned}$$

Therefore, applying Theorem 8.7 of [42] we conclude:

$$\mathbb{E}[\|\mathbb{P}_n - P\|_{\mathcal{K}}] \leq K_3 \left( \frac{\sigma}{\sqrt{n}} \sqrt{d \log \frac{K_1}{h^{d+1}C\sigma}} \vee \frac{dU}{n} \log \frac{K_1}{h^{d+1}C\sigma} \right)$$

$$\lesssim \left( \frac{1}{h^{d/2}\sqrt{n}} \sqrt{\log \frac{1}{h}} \vee \frac{1}{nh^d} \log \frac{1}{h} \right) = O\left( \sqrt{\frac{1}{nh^d}} \log \frac{1}{h} \right).$$

An application of Markov's inequality with the above bound on the expected value of the empirical process established the rate of equation (B.1).

For bound (B.2) we have:

$$\sup_{x \in \mathcal{X}} \left| \widetilde{d}_{n,h}(x) - \widetilde{\widetilde{d}}_{n,h}(x) \right|$$

$$= \sup_{x} \left| \frac{1}{n} \sum_{i=1}^{n} \frac{\frac{1}{h^d} K\left( \frac{\|x-X_i\|^2}{h^2} \right)}{\sqrt{d_{n,h}(x)}\sqrt{d_{n,h}(X_i)}} - \frac{1}{n} \sum_{i=1}^{n} \frac{\frac{1}{h^d} K\left( \frac{\|x-X_i\|^2}{h^2} \right)}{\sqrt{p_h(x)}\sqrt{p_h(X_i)}} \right|$$

$$\leq \sup_{x} \frac{1}{n} \sum_{i=1}^{n} \frac{1}{h^d} K\left( \frac{\|x-X_i\|^2}{h^2} \right) \left| \frac{1}{\sqrt{d_{n,h}(x)}\sqrt{d_{n,h}(X_i)}} - \frac{1}{\sqrt{p_h(x)}\sqrt{p_h(X_i)}} \right|$$

$$\leq \sup_{x,y \in \mathcal{X}} \left| \frac{1}{\sqrt{d_{n,h}(x)}\sqrt{d_{n,h}(y)}} - \frac{1}{\sqrt{p_h(x)}\sqrt{p_h(y)}} \right|$$
$$\times \sup_{x} \frac{1}{n} \sum_{i=1}^{n} \frac{1}{h^d} K\left( \frac{\|x-X_i\|^2}{h^2} \right)$$

$$\leq \sup_{x,y \in \mathcal{X}} \left| \frac{1}{\sqrt{d_{n,h}(x)}\sqrt{d_{n,h}(y)}} - \frac{1}{\sqrt{p_h(x)}\sqrt{p_h(y)}} \right|$$
$$\times \left[ \sup_{x} |\mathbb{P}_n K_{x,h} - P K_{x,h}| + \sup_{x} P K_{h,x} \right]$$

$$\leq \sup_{x,y \in \mathcal{X}} \left| \frac{1}{\sqrt{d_{n,h}(x)}\sqrt{d_{n,h}(y)}} - \frac{1}{\sqrt{p_h(x)}\sqrt{d_{n,h}(y)}} + \frac{1}{\sqrt{p_h(x)}\sqrt{d_{n,h}(y)}} - \frac{1}{\sqrt{p_h(x)}\sqrt{p_h(y)}} \right|$$
$$\times \left[ \sup_{x} |\mathbb{P}_n K_{x,h} - P K_{x,h}| + \sup_{x} P K_{h,x} \right]$$

$$\leq \left\{ \sup_{x \in \mathcal{X}} \frac{1}{\sqrt{d_{n,h}(x)}} \sup_{x} \left| \frac{1}{\sqrt{d_{n,h}(x)}} - \frac{1}{\sqrt{p_h(x)}} \right| + \sup_{x} \frac{1}{\sqrt{p_h(x)}} \sup_{x} \left| \frac{1}{\sqrt{d_{n,h}(y)}} - \frac{1}{\sqrt{p_h(y)}} \right| \right\}$$
$$\times \left[ \sup_{x} |\mathbb{P}_n K_{x,h} - P K_{x,h}| + \sup_{x} P K_{h,x} \right]$$

$$\leq \left\{ \sup_{x \in \mathcal{X}} \left| \frac{1}{\sqrt{d_{n,h}(x)}} - \frac{1}{\sqrt{p_h(x)}} \right| \sup_{x} \left| \frac{1}{\sqrt{d_{n,h}(x)}} - \frac{1}{\sqrt{p_h(x)}} \right| \right.$$
$$\left. + 2 \sup_{x} \frac{1}{\sqrt{p_h(x)}} \sup_{x} \left| \frac{1}{\sqrt{d_{n,h}(y)}} - \frac{1}{\sqrt{p_h(y)}} \right| \right\}$$
$$\times \left[ \sup_{x} |\mathbb{P}_n K_{x,h} - P K_{x,h}| + \sup_{x} P K_{h,x} \right]$$

$$\leq \left\{ \sup_{x} \left| \frac{1}{\sqrt{d_{n,h}(x)}} - \frac{1}{\sqrt{p_h(x)}} \right|^2 + 2 \sup_{x} \frac{1}{\sqrt{p_h(x)}} \sup_{x} \left| \frac{1}{\sqrt{d_{n,h}(y)}} - \frac{1}{\sqrt{p_h(y)}} \right| \right\}$$
$$\times \left[ \sup_{x} |\mathbb{P}_n K_{x,h} - P K_{x,h}| + \sup_{x} P K_{h,x} \right]$$

$$\leq \left\{ \sup_{x} \left| \frac{\sqrt{d_{n,h}(x)} - \sqrt{p_h(x)}}{\sqrt{d_{n,h}(x)p_h(x)}} \right|^2 + \frac{2}{\sqrt{\widetilde{p}_{\min}}} \sup_{x} \left| \frac{\sqrt{d_{n,h}(x)} - \sqrt{p_h(x)}}{\sqrt{d_{n,h}(x)p_h(x)}} \right| \right\}$$

$$\times \left[ \|\mathbb{P}_n - P\|_{\mathcal{K}} + \sup_x PK_{h,x} \right]$$

$$\leq \left\{ \sup_x \left| \frac{d_{n,h}(x) - p_h(x)}{\sqrt{d_{n,h}(x)p_h(x)} \left( \sqrt{d_{n,h}(x)} + \sqrt{p_h(x)} \right)} \right|^2 \right.$$

$$\left. + \frac{2}{\sqrt{\widetilde{p}_{\min}}} \sup_x \left| \frac{d_{n,h}(x) - p_h(x)}{\sqrt{d_{n,h}(x)p_h(x)} \left( \sqrt{d_{n,h}(x)} + \sqrt{p_h(x)} \right)} \right| \right\}$$

$$\times \left[ \|\mathbb{P}_n - P\|_{\mathcal{K}} + \sup_x PK_{h,x} \right] \tag{B.6}$$

$$= \left\{ O_p \left( \frac{1}{nh^d} \left( \log \frac{1}{h} \right)^2 \right) + O_p \left( \sqrt{\frac{1}{nh^d}} \log \frac{1}{h} \right) \right\} \times \left\{ O_p \left( \sqrt{\frac{1}{nh^d}} \log \frac{1}{h} \right) + O(1) \right\}$$

$$= O_p \left( \sqrt{\frac{1}{nh^d}} \log \frac{1}{h} \right).$$

where the rates follows from bound (B.1) along with the fact that the denominators of (B.6) are bounded away from 0. More precisely, the term $\left( \sqrt{d_{n,h}(x)} + \sqrt{p_h(x)} \right)$ in the denominator of equation (B.6) is lower bounded by $p_h(x)$ which is further uniformly lower bounded by $\widetilde{p}_{\min}$. To bound the other term $\sqrt{d_{n,h}(x)p_h(x)}$ in the denominator, we again use bound (B.1), from which we know for all $x \in \mathcal{X}$ and for all small $h$, we have $|d_{n,h}(x) - p_h(x)| \leq \widetilde{p}_{\min}/2$, which implies $d_{n,h} \geq p_{min}/2$. Therefore the term $\sqrt{d_{n,h}(x)p_h(x)}$ is lower bounded by $\widetilde{p}_{\min}/\sqrt{2}$. This completes the proof for bound (B.2).

The proof of bound (B.3) is similar to that of bound (B.1). Note that:

$$\sup_x \left| \widetilde{\widetilde{d}}_{n,h}(x) - \widetilde{\widetilde{d}}_h(x) \right|$$

$$= \sup_x \left| \frac{1}{n} \sum_{i=1}^n \frac{\frac{1}{h^d} K \left( \frac{\|x - X_i\|^2}{h^2} \right)}{\sqrt{p_h(x)} \sqrt{p_h(X_i)}} - \mathbb{E} \left[ \frac{\frac{1}{h^d} K \left( \frac{\|x - X\|^2}{h^2} \right)}{\sqrt{p_h(x)} \sqrt{p_h(X)}} \right] \right|$$

$$= \sup_x |\mathbb{P}_n g_x - P g_x|$$

where the function $g_x$ is defined as:

$$g_x(y) = \frac{\frac{1}{h^d} K \left( \frac{\|x - y\|^2}{h^2} \right)}{\sqrt{p_h(x)} \sqrt{p_h(y)}}.$$

Following the same line of argument as in the proof of bound (B.1) (with this new function class $\mathcal{G} = \{g_x : x \in \mathcal{X}\}$ instead of $\mathcal{K}$) we conclude the lemma. $\qquad \square$

We divide the rest of the proof into few steps. Henceforth we denote $\mathbb{L}_{n,srw}$ as $\mathbb{L}_n$ for typographical simplicity.

**Step 1:** Expanding the expression for $\mathbb{L}_n$ we have:

$$\mathbb{L}_n = \frac{1}{nh^2} f^\top \left( I - \widetilde{D}^{-1} \widetilde{K} \right) f$$

$$= \frac{1}{nh^2} \left[ \sum_i \left( 1 - \frac{\widetilde{K}_{ii}}{\widetilde{D}_{ii}} \right) f(X_i)^2 - \sum_{i \neq j} \frac{\widetilde{K}_{ij}}{\widetilde{D}_{ii}} f(X_i) f(X_j) \right]$$

We first show that the diagonal terms related to the scaled weighted matrix is asymptotically negligible, i.e.,

$$\frac{1}{nh^2} \sum_i \frac{\widetilde{K}_{ii}}{\widetilde{D}_{ii}} f^2(X_i) \xrightarrow{P} 0.$$

By the choice of our kernel, we have $\widetilde{K}_{ii} = 1/(nh^d d_{n,h}(X_i))$. Therefore we have:

$$\frac{1}{nh^2}\sum_i \frac{\widetilde{K}_{ii}}{\widetilde{D}_{ii}} f^2(X_i) = \frac{1}{nh^{d+2}} \times \frac{1}{n}\sum_i \frac{1}{d_{n,h}(X_i)\widetilde{d}_{n,h}(X_i)} f^2(X_i)$$

As per our assumption $nh^{d+2} \to \infty$, hence all we need to show is the second term in the above product is $O_p(1)$ to establish the claim. Towards that direction:

$$\left| \frac{1}{n}\sum_i \frac{f^2(X_i)}{d_{n,h}(X_i)\widetilde{d}_{n,h}(X_i)} \right|$$

$$\leq \frac{1}{n}\sum_i \frac{f^2(X_i)}{p_h(X_i)\widetilde{\widetilde{d}}_h(X_i)} + \frac{1}{n}\sum_i f^2(X_i)\left| \frac{1}{d_{n,h}(X_i)\widetilde{d}_{n,h}(X_i)} - \frac{1}{p_h(X_i)\widetilde{\widetilde{d}}_h(X_i)} \right|$$

That the first summand is $O_p(1)$ is immediate from the law of large numbers and the second term is $o_p(1)$ follows by a simple application of Lemma B.3 and Lemma B.2.

**Step 2:**   In the next step, we establish the following approximation of the off-diagonal terms:

$$\frac{1}{nh^2}\sum_{i\neq j} \frac{\widetilde{K}_{ij}}{\widetilde{D}_{ii}} f(X_i)f(X_j) = \frac{1}{nh^2}\sum_{i\neq j} \frac{\frac{1}{nh^d}K\left(\frac{\|X_i-X_j\|^2}{h^2}\right)}{\sqrt{p_h(X_i)}\sqrt{p_h(X_j)}} f(X_i)f(X_j) + o_p(1)\,.$$

We expand the difference as below:

$$\left| \frac{1}{nh^2}\sum_{i\neq j} \frac{\widetilde{K}_{ij}}{\widetilde{D}_{ii}} f(X_i)f(X_j) - \frac{1}{nh^2}\sum_{i\neq j} \frac{\frac{1}{nh^d}K\left(\frac{\|X_i-X_j\|^2}{h^2}\right)}{\sqrt{p_h(X_i)}\sqrt{p_h(X_j)}\widetilde{d}_h(X_i)} f(X_i)f(X_j) \right|$$

$$\leq \frac{1}{nh^2}\sum_{i\neq j}\left[ \frac{1}{nh^d}K\left(\frac{\|X_i-X_j\|^2}{h^2}\right) |f(X_i)f(X_j)| \times \right.$$

$$\left. \left| \frac{1}{\sqrt{d_{n,h}(X_i)d_{n,h}(X_j)}\widetilde{d}_{n,h}(X_i)} - \frac{1}{\sqrt{p_h(X_i)}\sqrt{p_h(X_j)}\widetilde{\widetilde{d}}_h(X_i)} \right| \right]$$

$$\leq \frac{1}{nh^2}\sum_{i\neq j} \frac{1}{nh^d}K\left(\frac{\|X_i-X_j\|^2}{h^2}\right) |f(X_i)f(X_j)| \times$$

$$\sup_{x,y}\left| \frac{1}{\sqrt{d_{n,h}(x)d_{n,h}(y)}\widetilde{d}_{n,h}(x)} - \frac{1}{\sqrt{p_h(x)}\sqrt{p_h(y)}\widetilde{\widetilde{d}}_h(x)} \right|$$

Again that the second term of the above product is $o_p(1)$ follows from the bounds established in Lemma B.3 and the lower bound in Lemma B.2. We now show that the first term of the above product in $O_p(1)$ which will conclude the claim.

$$\mathbb{E}\left[ \frac{1}{nh^2}\sum_{i\neq j} \frac{1}{nh^d}K\left(\frac{\|X_i-X_j\|^2}{h^2}\right) |f(X_i)f(X_j)| \right]$$

$$= \frac{1}{h^2}\mathbb{E}\left[ \frac{1}{h^d}K\left(\frac{\|X-Y\|^2}{h^2}\right) |f(X)f(Y)| \right]$$

$$= \frac{1}{h^2}\int_x\int_y \frac{1}{h^d}K\left(\frac{\|x-y\|^2}{h^2}\right) |f(x)f(y)|\, p(x)\, p(y)\, dx\, dy$$

$$\leq \frac{f_{\max}^2}{h^2}\int_x\int_y \frac{1}{h^d}K\left(\frac{\|x-y\|^2}{h^2}\right) p(x)\, p(y)\, dx\, dy$$

$$= \frac{f_{\max}^2}{h^2}\int_x p(x)\int_y \frac{1}{h^d}K\left(\frac{\|x-y\|^2}{h^2}\right) p(y)\, dy\, dx$$

$$= O\left(\frac{1}{h^2}\right).$$

Similar calculation as in the proof of Lemma B.3 we have:

$$\sup_{x,y}\left|\frac{1}{\sqrt{d_{n,h}(x)d_{n,h}(y)}\tilde{d}_{n,h}(x)} - \frac{1}{\sqrt{p_h(x)}\sqrt{p_h(y)}\tilde{\tilde{d}}_h(x)}\right| = O_p\left(\sqrt{\frac{1}{nh^d}\log\frac{1}{h}}\right)$$

Therefore we obtain:

$$\left|\frac{1}{nh^2}\sum_{i\neq j}\frac{\widetilde{K}_{ij}}{\widetilde{D}_{ii}}f(X_i)f(X_j) - \frac{1}{nh^2}\sum_{i\neq j}\frac{\frac{1}{nh^d}K\left(\frac{\|X_i-X_j\|^2}{h^2}\right)}{\sqrt{p_h(X_i)}\sqrt{p_h(X_j)}\tilde{\tilde{d}}_h(X_i)}f(X_i)f(X_j)\right|$$

$$\leq \frac{1}{nh^2}\sum_{i\neq j}\frac{1}{nh^d}K\left(\frac{\|X_i - X_j\|^2}{h^2}\right)|f(X_i)f(X_j)| \times$$

$$\sup_{x,y}\left|\frac{1}{\sqrt{d_{n,h}(x)d_{n,h}(y)}\tilde{d}_{n,h}(x)} - \frac{1}{\sqrt{p_h(x)}\sqrt{p_h(y)}\tilde{\tilde{d}}_h(x)}\right|$$

$$= O_p\left(\frac{1}{h^2}\right) \times O_p\left(\sqrt{\frac{1}{nh^d}\log\frac{1}{h}}\right)$$

$$= O_p\left(\sqrt{\frac{1}{nh^{d+4}}\log\frac{1}{h}}\right) = o_p(1).$$

$\square$

**Step 3:**    Based on our analysis in Step 1 and Step 2 we can write:

$$\mathbb{L}_n = \mathbb{L}_n^* + o_p(1)$$

where:

$$\mathbb{L}_n^* = \frac{1}{nh^2}\left[\sum_i f(X_i)^2 - \frac{1}{n}\sum_{i\neq j}\frac{\frac{K_h(\|X_i-X_j\|^2)}{\sqrt{p_h(X_i)p_h(X_j)}}}{\tilde{d}_h(X_i)}f(X_i)f(X_j)\right]$$

which can be further decomposed as the bias part and the variance part as follows:

$$\mathbb{L}_n^* = \underbrace{\mathbb{E}[\mathbb{L}_n^*]}_{Bias} + \underbrace{(\mathbb{L}_n^* - \mathbb{E}[\mathbb{L}_n^*])}_{Var}$$

In this step, we show that the variance part is $o_p(1)$. Towards that end, note that:

$$(\mathbb{L}_n^* - \mathbb{E}[\mathbb{L}_n^*])$$

$$= \frac{1}{h^2}\left[\frac{1}{n}\sum_{i=1}^n h^2(X_i) - \mathbb{E}[f(X)]\right]$$

$$+ \frac{1}{h^2}\left[\frac{1}{n^2}\sum_{i\neq j}\frac{\frac{K_h(\|X_i-X_j\|^2)}{\sqrt{p_h(X_i)p_h(X_j)}}}{\tilde{d}_h(X_i)}f(X_i)f(X_j) - \frac{(n-1)}{n}\mathbb{E}\left[\frac{K_h(\|X-Y\|^2)}{\tilde{d}_h(X)\sqrt{p_h(X)p_h(Y)}}f(X)f(Y)\right]\right]$$

$$= O_p\left(\frac{1}{h^2\sqrt{n}}\right) = o_p(1).$$

where the rate in the last line follows from the fact that:

$$\text{var}(f(X)) = O(1) \quad \text{and}$$

$$\text{var}\left(\frac{K_h(\|X-Y\|^2)}{\tilde{d}_h(X)\sqrt{p_h(X)p_h(Y)}}f(X)f(Y)\right) = O(1).$$

**Step 4:** In the last step of the proof we show that:

$$\mathbb{E}[\mathbb{L}_n^*] \xrightarrow{P} \mathbb{E}[\|\nabla f(X)\|^2].$$

Towards that end, note that:

$$\mathbb{E}[\mathbb{L}_n^*] = \frac{1}{h^2}\left[\int f^2(x)p(x)\,dx - \int\int \frac{K\left(\frac{\|x-y\|^2}{h^2}\right)}{h^d \widetilde{d}_h(x)\sqrt{p_h(x)p_h(y)}}f(x)f(y)\,p(x)p(y)\,dxdy\right]$$

We will use DCT to establish the convergence of the above integral. The above integral can be written as:

$$\mathbb{E}[\mathbb{L}_n^*] = \int_{\mathcal{X}} g_h(x)\,p(x)\,dx$$

where the function $g_h(x) \equiv g_{h_n}(x)$ is defined as:

$$
\begin{aligned}
g_h(x) &= \frac{1}{h^2}\left[f^2(x) - f(x)\int_{\mathcal{X}} \frac{K\left(\frac{\|x-y\|^2}{h^2}\right)}{h^d \widetilde{d}_h(x)\sqrt{p_h(x)p_h(y)}}f(y)\,p(y)\,dy\right] \\
&:= \frac{1}{h^2}\left[f^2(x) - f(x)\int_{\mathcal{X}} f(y)m_{h,x}(y)\,dy\right]
\end{aligned}
\tag{B.7}
$$

with the *transformed probability density function* $m_{h,x}(\cdot)$ is defined as:

$$m_{h,x}(y) = \frac{K_h(\|x-y\|^2)\frac{p(y)}{\sqrt{p_h(y)}}}{\int_{\mathcal{X}} K_h(\|x-y\|^2)\frac{p(y)}{\sqrt{p_h(y)}}\,dy} = \frac{K_h(\|x-y\|^2)\frac{p(y)}{\sqrt{p_h(y)}}}{\Lambda(x)}.$$

It is proved in [25] (see main result in Section 3.3) that this sequence of functions convergence pointwise to the range of Laplacian operator, i.e.,

$$g_h(x) \xrightarrow{\text{ptwise}} -f(x)\Delta f(x)$$

which ensures the convergence in probability of the sequence of random variables $\{g_{h_n}(X)\}$. Therefore if we can show that the sequence $\{g_{h_n}(X)\}$ is uniformly integrable, i.e there exists some $\delta > 0$ such that:

$$\limsup_n \mathbb{E}\left[|g_{h_n}(X)|^{1+\delta}\right] < \infty$$

then an application of Vitali's theorem yields $L_1$ convergence, i.e.,

$$\mathbb{E}\left[g_{h_n}(X)\right] \to \mathbb{E}\left[-f(X)\Delta f(X)\right] = \mathbb{E}\left[\|\nabla f(X)\|^2\right]$$

where the last equality is obtained by applying Green's theorem. Therefore all we need to do is to establish uniform integrability of the sequence $\{g_{h_n}(X)\}$. Note that, a two step Taylor expansion of equation (B.7) yields:

$$
\begin{aligned}
&\left|\frac{1}{h^2}\left[f^2(x) - f(x)\int_{\mathcal{X}} f(y)m_{h,x}(y)\,dy\right]\right| \\
&= \left|\frac{1}{h^2}\left[\nabla f(x)^\top \int_{\mathcal{X}}(y-x)m_{h,x}(y)\,dy + \int_{\mathcal{X}}\frac{1}{2}(y-x)^\top\nabla^2 f(\widetilde{y})(y-x)\,m_{h,x}(y)\,dy\right]\right| \\
&\leq \left|\frac{1}{h}\frac{\nabla f(x)^\top}{\Lambda(x)}\int_{\frac{\mathcal{X}-x}{h}} zK(\|z\|^2)\frac{p(x+hz)}{\sqrt{p_h(x+hz)}}\,dy\right| \\
&\qquad + \sup_x \|\nabla^2 f(x)\|_{op} \times \frac{1}{\Lambda(x)}\int_{\frac{\mathcal{X}-x}{h}} \|z\|^2 K(\|z\|^2)\frac{p(x+hz)}{\sqrt{p_h(x+hz)}}\,dy \\
&= T_1 + T_2
\end{aligned}
$$

Bound $T_2$ is easier, as we have already established in Lemma B.2 that $p_h(x)$ is uniformly lower bounded on $\mathcal{X}$ and $p(x)$ is uniformly upper bounded by our assumption. We now show that the lower bound on $p_h(x)$ translates to the lower bound on $\Lambda(x)$ as:

$$
\begin{aligned}
\Lambda(x) &= \int_{\mathcal{X}} K_h(\|x - y\|^2) \frac{p(y)}{\sqrt{p_h(y)}} \, dy \\
&= \int_{\frac{\mathcal{X}-x}{h}} K(\|z\|^2) \frac{p(x + hz)}{\sqrt{p_h(x + hz)}} \, dy \\
&\geq \frac{p_{\min}}{p_{\max}} \int_{\frac{\mathcal{X}-x}{h}} K(\|z\|^2) \, dz \\
&\geq \frac{p_{\min}}{p_{\max}} \times \inf_{x \in \mathcal{X}} \int_{\frac{\mathcal{X}-x}{h}} K(\|z\|^2) \, dz := \widetilde{\Lambda}.
\end{aligned}
$$

This implies that $T_2$ is upper bounded by a constant as:

$$
T_2 \leq \sup_x \|\nabla^2 f(x)\|_{op} \times \frac{p_{\max}}{\widetilde{p}_{\min}\widetilde{\Lambda}} \int_{\mathbb{R}^d} \|z\|^2 K(\|z\|^2) \, dy.
$$

Bounding $T_1$ is a bit more tricky. First we have:

$$
\begin{aligned}
&\frac{1}{h} \int_{\frac{\mathcal{X}-x}{h}} \nabla f(x)^\top z K(\|z\|^2) \frac{p(x + hz)}{\Lambda(x)\sqrt{p_h(x + hz)}} \, dy \\
&= \frac{1}{h} \int_{\frac{\mathcal{X}-x}{h}} \nabla f(x)^\top z K(\|z\|^2) \, dz \\
&\quad + \frac{1}{h} \int_{\frac{\mathcal{X}-x}{h}} \nabla f(x)^\top z K(\|z\|^2) \left( \frac{p(x + hz)}{\Lambda(x)\sqrt{p_h(x + hz)}} - 1 \right) \, dz \\
&= \frac{1}{h} \int_{\left(\frac{\mathcal{X}-x}{h}\right)^c} \nabla f(x)^\top z K(\|z\|^2) \, dz \\
&\quad + \frac{1}{h} \int_{\frac{\mathcal{X}-x}{h}} \nabla f(x)^\top z K(\|z\|^2) \left( \frac{p(x + hz)}{\Lambda(x)\sqrt{p_h(x + hz)}} - 1 \right) \, dz \\
&= \frac{1}{h} \int_{\left(\frac{\mathcal{X}-x}{h}\right)^c} \nabla f(x)^\top z K(\|z\|^2) \, dz \\
&\quad + \frac{1}{h\Lambda(x)} \int_{\frac{\mathcal{X}-x}{h}} \nabla f(x)^\top z K(\|z\|^2) \left( \frac{p(x + hz)}{\sqrt{p_h(x + hz)}} - \Lambda(x) \right) \, dz \\
&= \frac{1}{h} \int_{\left(\frac{\mathcal{X}-x}{h}\right)^c} \nabla f(x)^\top z K(\|z\|^2) \, dz \\
&\quad + \frac{1}{h\Lambda(x)} \int_{\frac{\mathcal{X}-x}{h}} \nabla f(x)^\top z K(\|z\|^2) \left( \frac{p(x + hz)}{\sqrt{p_h(x + hz)}} - \int_{\mathcal{X}} K_h(\|x - y\|^2) \frac{p(y)}{\sqrt{p_h(y)}} \, dy \right) \, dz \\
&= \frac{1}{h} \int_{\left(\frac{\mathcal{X}-x}{h}\right)^c} \nabla f(x)^\top z K(\|z\|^2) \, dz \\
&\quad + \frac{1}{h\Lambda(x)} \int_{\frac{\mathcal{X}-x}{h}} \nabla f(x)^\top z K(\|z\|^2) \left( \frac{p(x + hz)}{\sqrt{p_h(x + hz)}} - \int_{\frac{\mathcal{X}-x}{h}} K_h(\|w\|^2) \frac{p(x + hw)}{\sqrt{p_h(x + hw)}} \, dy \right) \, dw \\
&= \frac{1}{h} \int_{\left(\frac{\mathcal{X}-x}{h}\right)^c} \nabla f(x)^\top z K(\|z\|^2) \, dz \\
&\quad + \frac{1}{h\Lambda(x)} \int_{\frac{\mathcal{X}-x}{h}} \nabla f(x)^\top z K(\|z\|^2) \left( \frac{p(x + hz)}{\sqrt{p_h(x + hz)}} \left( 1 - \int_{\frac{\mathcal{X}-x}{h}} K_h(\|w\|^2) \, dw \right) \right) \, dz \\
&\quad - \frac{1}{h\Lambda(x)} \int_{\frac{\mathcal{X}-x}{h}} \nabla f(x)^\top z K(\|z\|^2) \int_{\frac{\mathcal{X}-x}{h}} K_h(\|w\|^2) \left( \frac{p(x + hw)}{\sqrt{p_h(x + hw)}} - \frac{p(x + hz)}{\sqrt{p_h(x + hz)}} \right) \, dw
\end{aligned}
$$

$$= \frac{1}{h} \int_{\left(\frac{\mathcal{X}-x}{h}\right)^c} \nabla f(x)^\top z K(\|z\|^2) \, dz$$

$$+ \frac{1}{h\Lambda(x)} \int_{\left(\frac{\mathcal{X}-x}{h}\right)^c} K_h(\|w\|^2) \, dw \int_{\frac{\mathcal{X}-x}{h}} \nabla f(x)^\top z K(\|z\|^2) \frac{p(x+hz)}{\sqrt{p_h(x+hz)}} \, dz$$

$$- \frac{1}{\Lambda(x)} \int_{\frac{\mathcal{X}-x}{h}} \nabla f(x)^\top z K(\|z\|^2) \left( \int_{\frac{\mathcal{X}-x}{h}} K(\|w\|^2) \left\langle w - z, \nabla \left( \frac{p}{\sqrt{p_h}} \right)(\tilde{w}_{x,z}) \right\rangle \, dw \right) dz$$

$$\text{(B.8)}$$

Note that the value $\tilde{w}_{x,z}$ is an intermediate value between $x + hz$ and $x + hw$ which can be written as $\tilde{w}_{x,z} = x + h(\alpha z + (1-\alpha)w)$ for some $\alpha \in [0,1]$ depending on $x, z, w$. The gradient of $p_h(x)$ is:

$$\nabla p_h(x) = \frac{d}{dx} \int_{\mathbb{R}^d} \frac{1}{(2\pi)^{d/2} h^d} e^{-\frac{1}{2h^2}\|x-y\|^2} \mathbb{1}_{y\in\mathcal{X}} \, p(y) \, dy$$

$$= \frac{1}{2h} \int_{\mathbb{R}^d} \frac{y-x}{h} \frac{1}{(2\pi)^{d/2} h^d} e^{-\frac{1}{2h^2}\|x-y\|^2} \mathbb{1}_{y\in\mathcal{X}} \, p(y) \, dy$$

$$= \frac{1}{2h} \int_{\mathbb{R}^d} y K(\|y\|^2) \mathbb{1}_{y\in\frac{\mathcal{X}-x}{h}} \, p(x+hy) \, dy$$

$$= \frac{p(x)}{2h} \int_{\mathbb{R}^d} y K(\|y\|^2) \mathbb{1}_{y\in\frac{\mathcal{X}-x}{h}} \, dy + O(1)$$

$$:= p(x)g(x) + O(1)$$

where the $O(1)$ term is uniform over the entire region $\mathcal{X}$. For the entire thing $p/\sqrt{p_h}$:

$$\nabla \left( \frac{p}{\sqrt{p_h}} \right)(x) = p_h^{-1/2}(x)\nabla p(x) - \frac{1}{2}p(x)p_h(x)^{-3/2}\nabla p_h(x)$$

$$= \frac{1}{2}p^2(x)p_h(x)^{-3/2}g(x) + R(x)$$

where the remainder term $R(x)$ is uniformly bounded over $\mathcal{X}$. Now going back to equation (B.8) we have:

$$T_1 = \frac{1}{h} \int_{\left(\frac{\mathcal{X}-x}{h}\right)^c} \nabla f(x)^\top z K(\|z\|^2) \, dz$$

$$+ \frac{1}{h\Lambda(x)} \int_{\left(\frac{\mathcal{X}-x}{h}\right)^c} K_h(\|w\|^2) \, dw \int_{\frac{\mathcal{X}-x}{h}} \nabla f(x)^\top z K(\|z\|^2) \frac{p(x+hz)}{\sqrt{p_h(x+hz)}} \, dz$$

$$- \frac{1}{\Lambda(x)} \int_{\frac{\mathcal{X}-x}{h}} \nabla f(x)^\top z K(\|z\|^2) \left( \int_{\frac{\mathcal{X}-x}{h}} K(\|w\|^2) \left\langle w - z, \nabla \left( \frac{p}{\sqrt{p_h}} \right)(\tilde{w}_{x,z}) \right\rangle \, dw \right) dz$$

$$= \frac{1}{h} \int_{\left(\frac{\mathcal{X}-x}{h}\right)^c} \nabla f(x)^\top z K(\|z\|^2) \, dz$$

$$+ \frac{1}{h\Lambda(x)} \int_{\left(\frac{\mathcal{X}-x}{h}\right)^c} K_h(\|w\|^2) \, dw \int_{\frac{\mathcal{X}-x}{h}} \nabla f(x)^\top z K(\|z\|^2) \frac{p(x+hz)}{\sqrt{p_h(x+hz)}} \, dz$$

$$+ \frac{1}{\Lambda(x)} \int_{\frac{\mathcal{X}-x}{h}} \nabla f(x)^\top z K(\|z\|^2) \left( \int_{\frac{\mathcal{X}-x}{h}} K(\|w\|^2) \left\langle w - z, \frac{1}{2}p^2(\tilde{w}_{x,z})p_h(\tilde{w}_{x,z})^{-3/2}g(\tilde{w}_{x,z})) \right\rangle \, dw \right) dz$$

$$- \frac{1}{\Lambda(x)} \int_{\frac{\mathcal{X}-x}{h}} \nabla f(x)^\top z K(\|z\|^2) \left( \int_{\frac{\mathcal{X}-x}{h}} K(\|w\|^2) \left\langle w - z, R(\tilde{w}_{x,z}) \right\rangle \, dw \right) dz$$

$$= \frac{1}{h} \int_{\left(\frac{\mathcal{X}-x}{h}\right)^c} \nabla f(x)^\top z K(\|z\|^2) \, dz$$

$$+ \frac{1}{h\Lambda(x)} \int_{\left(\frac{\mathcal{X}-x}{h}\right)^c} K_h(\|w\|^2) \, dw \int_{\frac{\mathcal{X}-x}{h}} \nabla f(x)^\top z K(\|z\|^2) \frac{p(x+hz)}{\sqrt{p_h(x+hz)}} \, dz$$

$$+ \frac{1}{\Lambda(x)} \int_{\frac{\mathcal{X}-x}{h}} \nabla f(x)^\top z K(\|z\|^2) \left( \int_{\frac{\mathcal{X}-x}{h}} K(\|w\|^2) \left\langle w - z, \frac{1}{2} p^2(\widetilde{w}_{x,z}) p_h(\widetilde{w}_{x,z})^{-3/2} g(x) \right\rangle dw \right) dz$$

$$+ \frac{1}{\Lambda(x)} \int_{\frac{\mathcal{X}-x}{h}} \nabla f(x)^\top z K(\|z\|^2) \left( \int_{\frac{\mathcal{X}-x}{h}} K(\|w\|^2) \left\langle w - z, \left( \frac{1}{2} p^2(\widetilde{w}_{x,z}) p_h(\widetilde{w}_{x,z})^{-3/2} g(\widetilde{w}_{x,z}) \right) \right. \right.$$

$$\left. \left. - \frac{1}{2} p^2(\widetilde{w}_{x,z}) p_h(\widetilde{w}_{x,z})^{-3/2} g(x) \right) \right\rangle dw \right) dz$$

$$- \frac{1}{\Lambda(x)} \int_{\frac{\mathcal{X}-x}{h}} \nabla f(x)^\top z K(\|z\|^2) \left( \int_{\frac{\mathcal{X}-x}{h}} K(\|w\|^2) \langle w - z, R(\widetilde{w}_{x,z}) \rangle \, dw \right) dz$$

$$:= T_{11} + T_{12} + T_{13} + T_{14} + T_{15}$$

.

Therefore the function $g_h(x)$ is bouned by:

$$|g_{h_n}(x)| \leq T_{11,n} + T_{12,n} + T_{13,n} + T_{14,n} + T_{15,n} + T_{2,n}. \tag{B.9}$$

We next prove that the collection of functions $\{g_{h_n}(x)\}_n$ is uniformly integrable, for which it is enough to show that each term on the above bound of $|g_h|$ is uniformly integrable. We have already established in equation (B.1.2) that $T_2$ is uniformly bounded by a constant hence U.I. As for $T_{15}$, we already know that the function $R(x)$ is uniformly bounded, which immediately implies:

$$|T_{15,n}(x)| = \left| \frac{1}{\Lambda(x)} \int_{\frac{\mathcal{X}-x}{h}} \nabla f(x)^\top z K(\|z\|^2) \left( \int_{\frac{\mathcal{X}-x}{h}} K(\|w\|^2) \langle w - z, R(\widetilde{w}_{x,z}) \rangle \, dw \right) dz \right|$$

$$\leq \left( \sup_{z \in \mathcal{X}} \|R(z)\| \right) \times \|\nabla f(x)\| \int_{\frac{\mathcal{X}-x}{h}} \|z\| K(\|z\|^2) \int_{\frac{\mathcal{X}-x}{h}} \|w - z\| K(\|w\|^2) \, dw \, dz$$

$$\leq C_5.$$

and consequently $\{T_{15,n}\}$ is U.I. The other parts need a more involved calculations. We start with $T_{11,n}$:

$$\mathbb{E}[T_{11,n}^{1+\delta}] = \mathbb{E}_X \left[ \left( \|\nabla f(x)\| \times \frac{1}{h} \int_{\left( \frac{\mathcal{X}-x}{h} \right)^c} \|z\| K(\|z\|^2) \, dz \right)^{1+\delta} \right]$$

$$= \mathbb{E}_X \left[ \left( \|\nabla f(x)\| \times \frac{1}{h} \mathbb{E}_Z \left[ \|Z\| \mathbb{1}_{X+Zh \notin \mathcal{X}} \right] \right)^{1+\delta} \right]$$

$$= \mathbb{E}_X \left[ \left( \|\nabla f(x)\| \times \frac{1}{h} \mathbb{E}_Z \left[ \|Z\| \mathbb{1}_{X+Zh \notin \mathcal{X}, \|Z\| \leq 2\sqrt{\log \frac{1}{h}}} \right] \right)^{1+\delta} \right]$$

$$+ \mathbb{E}_X \left[ \left( \|\nabla f(x)\| \times \frac{1}{h} \mathbb{E}_Z \left[ \|Z\| \mathbb{1}_{X+Zh \notin \mathcal{X}, \|Z\| > 2\sqrt{\log \frac{1}{h}}} \right] \right)^{1+\delta} \right]$$

$$\leq \mathbb{E}_X \left[ \left( \|\nabla f(x)\| \times \frac{1}{h} \mathbb{E}_Z \left[ \|Z\| \mathbb{1}_{X+Zh \notin \mathcal{X}, \|Z\| \leq 2\sqrt{\log \frac{1}{h}}} \right] \right)^{1+\delta} \right]$$

$$+ \mathbb{E}_X \left[ \left( \|\nabla f(x)\| \times \frac{1}{h} \mathbb{E}_Z \left[ \|Z\| \mathbb{1}_{\|Z\| > 2\sqrt{\log \frac{1}{h}}} \right] \right)^{1+\delta} \right]$$

$$\leq \mathbb{E}_X \left[ \left( \|\nabla f(x)\| \times \frac{1}{h} \mathbb{E}_Z \left[ \|Z\| \mathbb{1}_{X+Zh \notin \mathcal{X}, \|Z\| \leq 2\sqrt{\log \frac{1}{h}}} \right] \right)^{1+\delta} \right]$$

$$+ \mathbb{E}_X \left[ \left( \|\nabla f(x)\| \times \frac{1}{h} e^{-\log \frac{1}{h}} \right)^{1+\delta} \right]$$

$$\leq \mathbb{E}_X \left[ \left( \|\nabla f(x)\| \times \frac{1}{h} \mathbb{E}_Z \left[ \|Z\| \mathbb{1}_{X+Zh \notin \mathcal{X}, \|Z\| \leq 2\sqrt{\log \frac{1}{h}}} \right] \right)^{1+\delta} \right] + C_1$$

$$
= \mathbb{E}_X \left[ \left( \mathbb{E}_Z \left[ \frac{\|\nabla f(x)\|}{h} \|Z\| \mathbb{1}_{X+Zh \notin \mathcal{X}, \|Z\| \leq 2\sqrt{\log \frac{1}{h}}} \right] \right)^{1+\delta} \right] + C_1
$$

$$
\leq \mathbb{E}_X \left[ \left( \frac{\|\nabla f(x)\|}{h} \right)^{1+\delta} \mathbb{E}_Z \left[ \|Z\|^{1+\delta} \mathbb{1}_{X+Zh \notin \mathcal{X}, \|Z\| \leq 2\sqrt{\log \frac{1}{h}}} \right] \right] + C_1
$$

$$
= \mathbb{E}_Z \left[ \frac{\|Z\|^{1+\delta}}{h} \mathbb{E}_X \left[ \left( \frac{\|\nabla f(x)\|^{1+\delta}}{h^\delta} \right) \mathbb{1}_{X+Zh \notin \mathcal{X}, \|Z\| \leq 2\sqrt{\log \frac{1}{h}}} \right] \right] + C_1
$$

$$
\leq \sup_{x:d_{b,\mathcal{X}}(x) \leq h\sqrt{\log \frac{1}{h}}} \left( \frac{\|\nabla f(x)\|^{1+\delta}}{h^\delta} \right) \times \mathbb{E}_Z \left[ \frac{\|Z\|^{1+\delta}}{h} \mathbb{E}_X \left[ \mathbb{1}_{X+Zh \notin \mathcal{X}, \|Z\| \leq 2\sqrt{\log \frac{1}{h}}} \right] \right] + C_1
$$

$$
= \sup_{x:d_{b,\mathcal{X}}(x) \leq h\sqrt{\log \frac{1}{h}}} \left( \frac{\|\nabla f(x)\|^{1+\delta}}{h^\delta} \right) \times \mathbb{E}_Z \left[ \frac{\|Z\|^{1+\delta}}{h} P\left(X + Zh \notin \mathcal{X}\right) \mathbb{1}_{\|Z\| \leq 2\sqrt{\log \frac{1}{h}}} \right] + C_1
$$

$$
= \sup_{x:d_{b,\mathcal{X}}(x) \leq h\sqrt{\log \frac{1}{h}}} \left( \frac{\|\nabla f(x)\|^{1+\delta}}{h^\delta} \right) \times \mathbb{E}_Z \left[ \frac{\|Z\|^{1+\delta}}{h} h\|Z\| \frac{P\left(X + Zh \notin \mathcal{X}\right)}{h\|Z\|} \mathbb{1}_{h\|Z\| \leq 2h\sqrt{\log \frac{1}{h}}} \right] + C_1
$$

$$
\leq \sup_{x:d_{b,\mathcal{X}}(x) \leq h\sqrt{\log \frac{1}{h}}} \left( \frac{\|\nabla f(x)\|^{1+\delta}}{h^\delta} \right) \times \sup_{\|t\| \leq 2h\sqrt{\log \frac{1}{h}}} \frac{P\left(X + t \notin \mathcal{X}\right)}{t} \times \mathbb{E}[\|Z\|^{2+\delta}] + C_1
$$

$$
\leq C_1 + C_2 \,.
$$

Therefore we can establish the sequence $\{T_{11,n}\}$ is U.I. provided that:

$$
\sup_{x:d_{b,\mathcal{X}}(x) \leq h\sqrt{\log \frac{1}{h}}} \left( \frac{\|\nabla f(x)\|^{1+\delta}}{h^\delta} \right) = O(1)
$$

for some $\delta > 0$ and

$$
\sup_{\|t\| \leq 2h\sqrt{\log \frac{1}{h}}} \frac{P\left(X + t \notin \mathcal{X}\right)}{\|t\|} = O(1) \,.
$$

The first condition follows immediately from our assumption $\nabla f(x) = 0$ at the boundary of $\mathcal{X}$ and the second condition follows from our assumption $p(x)$ is uniformly lower bounded on $\mathcal{X}$. To show that the other sequence $\{T_{12,n}\}$ is U.I. fix a small $\delta > 0$ constant $L$ such that $L^2 \geq 2(1 + \delta)$. We have:

$$
\mathbb{E}_X[|T_{12,n}^{1+\delta}|] = \mathbb{E}_X \left[ \left| \frac{1}{h\Lambda(x)} \int_{\left(\frac{\mathcal{X}-x}{h}\right)^c} K_h(\|w\|^2) \, dw \int_{\frac{\mathcal{X}-x}{h}} \nabla f(x)^\top z K(\|z\|^2) \frac{p(x+hz)}{\sqrt{p_h(x+hz)}} \, dz \right|^{1+\delta} \right]
$$

$$
\leq \left( \frac{p_{\max}}{\widetilde{\Lambda}\widetilde{p}_{\min}} \right)^{1+\delta} \mathbb{E}_X \left[ \frac{\|\nabla f(x)\|^{1+\delta}}{h^{1+\delta}} \left( \mathbb{E}_Z \left[ \mathbb{1}_{X+hZ \notin \mathcal{X}} \right] \right)^{1+\delta} \right]
$$

$$
\leq \left( \frac{p_{\max}}{\widetilde{\Lambda}\widetilde{p}_{\min}} \right)^{1+\delta} \mathbb{E}_X \left[ \frac{\|\nabla f(x)\|^{1+\delta}}{h^{1+\delta}} \mathbb{E}_Z \left[ \mathbb{1}_{X+hZ \notin \mathcal{X}} \right] \right] \qquad \text{[Jensen's inequality]}
$$

$$
= \left( \frac{p_{\max}}{\widetilde{\Lambda}\widetilde{p}_{\min}} \right)^{1+\delta} \left[ \mathbb{E}_X \left[ \frac{\|\nabla f(x)\|^{1+\delta}}{h^{1+\delta}} \mathbb{E}_Z \left[ \mathbb{1}_{X+hZ \notin \mathcal{X}, \, \|Z\| \leq L\sqrt{\log \frac{1}{h}}} \right] \right] \right.
$$

$$
\left. + \mathbb{E}_X \left[ \frac{\|\nabla f(x)\|^{1+\delta}}{h^{1+\delta}} \mathbb{E}_Z \left[ \mathbb{1}_{X+hZ \notin \mathcal{X}, \, \|Z\| > L\sqrt{\log \frac{1}{h}}} \right] \right] \right]
$$

$$
\leq \left( \frac{p_{\max}}{\widetilde{\Lambda}\widetilde{p}_{\min}} \right)^{1+\delta} \left[ \mathbb{E}_X \left[ \frac{\|\nabla f(x)\|^{1+\delta}}{h^{1+\delta}} \mathbb{E}_Z \left[ \mathbb{1}_{X+hZ \notin \mathcal{X}, \, \|Z\| \leq L\sqrt{\log \frac{1}{h}}} \right] \right] \right.
$$

$$
\left. + \mathbb{E}_X \left[ \frac{\|\nabla f(x)\|^{1+\delta}}{h^{1+\delta}} \mathbb{E}_Z \left[ \mathbb{1}_{\|Z\| > L\sqrt{\log \frac{1}{h}}} \right] \right] \right]
$$

$$
\leq \left( \frac{p_{\max}}{\widetilde{\Lambda}\widetilde{p}_{\min}} \right)^{1+\delta} \left[ \mathbb{E}_Z \left[ \mathbb{E}_X \left[ \frac{\|\nabla f(X)\|^{1+\delta}}{h^{1+\delta}} \mathbb{1}_{X+hZ \notin \mathcal{X}} \mathbb{1}_{\|Z\| \leq L\sqrt{\log \frac{1}{h}}} \right] \right] \right]
$$

$$+ \frac{h^{L^2/2}}{h^{1+\delta}} \mathbb{E}_X \left[ \|\nabla f(X)\|^{1+\delta} \right] \Bigg]$$

$$\leq \left( \frac{p_{\max}}{\widetilde{\Lambda} \widetilde{p}_{\min}} \right)^{1+\delta} \times \sup_{x: d_b(x) \leq Lh\sqrt{1/h}} \frac{\|\nabla f(x)\|^{1+\delta}}{h^\delta}$$

$$\times \mathbb{E}_Z \left[ \frac{1}{h} \mathbb{P}_X \left( X + hZ \notin \mathcal{X} \right) \mathbb{1}_{\|Z\| \leq L\sqrt{\log \frac{1}{h}}} \right] + O(1)$$

$$\leq \left( \frac{p_{\max}}{\widetilde{\Lambda} \widetilde{p}_{\min}} \right)^{1+\delta} \times \sup_{x: d_b(x) \leq Lh\sqrt{1/h}} \frac{\|\nabla f(x)\|^{1+\delta}}{h^\delta}$$

$$\times \sup_{\|t\| \leq Lh\sqrt{\log \frac{1}{h}}} \frac{P\left( X + t \notin \mathcal{X} \right)}{\|t\|} \times \mathbb{E} \left[ \|Z\| \mathbb{1}_{\|Z\| \leq L\sqrt{\log \frac{1}{h}}} \right] + O(1)$$

$$\leq C_{12}.$$

Now for $\{T_{13,n}\}$:

$$\mathbb{E}[|T_{13,n}|^{1+\delta}] = \mathbb{E} \left[ \left| \frac{1}{\Lambda(x)} \int_{\frac{\mathcal{X}-x}{h}} \nabla f(x)^\top z K(\|z\|^2) \left( \int_{\frac{\mathcal{X}-x}{h}} K(\|w\|^2) \langle w - z, \right. \right. \right.$$

$$\left. \left. \left. \frac{1}{2} p^2(\widetilde{w}_{x,z}) p_h(\widetilde{w}_{x,z})^{-3/2} g(x)) \rangle \, dw \right) dz \right|^{1+\delta} \right]$$

$$= \left( \frac{p_{\max}^2}{\widetilde{\Lambda} \widetilde{p}_{\min}^{3/2}} \int_{\mathbb{R}^d} \int_{\mathbb{R}^d} \|z\| \|w - z\| K(\|z\|^2) K(\|w\|^2) \, dz dw \right)^{1+\delta}$$

$$\times \mathbb{E}_X \left[ \|\nabla f(X)\|^{1+\delta} \|g(X)\|^{1+\delta} \right]$$

$$= \left( \frac{p_{\max}^2}{\widetilde{\Lambda} \widetilde{p}_{\min}^{3/2}} \int_{\mathbb{R}^d} \int_{\mathbb{R}^d} \|z\| \|w - z\| K(\|z\|^2) K(\|w\|^2) \, dz dw \right)^{1+\delta}$$

$$\times \mathbb{E} \left[ \left\| \frac{\|\nabla f(X)\|}{2h} \int_{\mathbb{R}^d} y K(\|y\|^2) \mathbb{1}_{y \in \frac{\mathcal{X}-x}{h}} \, dy \right\|^{1+\delta} \right]$$

As can be seen the rest of the proof is analogous to that of bounding $\mathbb{E} \left[ |T_{11,n}|^{1+\delta} \right]$ and hence skipped for brevity. Finally for showing $\{T_{14,n}\}$ is U.I.:

$$\mathbb{E}[|T_{14,n}|^{1+\delta}] = \mathbb{E} \left[ \left| \frac{1}{\Lambda(x)} \int_{\frac{\mathcal{X}-x}{h}} \nabla f(x)^\top z K(\|z\|^2) \left( \int_{\frac{\mathcal{X}-x}{h}} K(\|w\|^2) \left\langle w - z, \left( \frac{1}{2} p^2(\widetilde{w}_{x,z}) p_h(\widetilde{w}_{x,z})^{-3/2} g(\widetilde{w}_{x,z}) \right) \right. \right. \right. \right.$$

$$\left. \left. \left. \left. - \frac{1}{2} p^2(\widetilde{w}_{x,z}) p_h(\widetilde{w}_{x,z})^{-3/2} g(x)) \right\rangle \, dw \right) dz \right|^{1+\delta} \right]$$

$$\leq \left( \frac{p_{\max}}{2\widetilde{\Lambda} \widetilde{p}_{\min}^{3/2}} \right)^{1+\delta} \mathbb{E}_X \left[ \left| \int_{\frac{\mathcal{X}-x}{h}} \int_{\frac{\mathcal{X}-x}{h}} \|\nabla f(X)\| \|z\| \|w - z\| K(\|z\|^2) K(\|w\|^2) \|g(\widetilde{w}_{x,z})) - g(x)\| \right|^{1+\delta} \right]$$

$$\leq \left( \frac{p_{\max}}{2\widetilde{\Lambda} \widetilde{p}_{\min}^{3/2}} \right)^{1+\delta} \mathbb{E}_X \left[ \|\nabla f(X)\|^{1+\delta} \left| \mathbb{E}_{Z_1, Z_2} \left[ \|Z_1\| \|Z_1 - Z_2\| \mathbb{E}_{Z_3} \left[ \frac{1}{2h} \|Z_3\| \left( \mathbb{1}_{Z_3 \in \left( \frac{\mathcal{X}-x}{h} \right)^c} \right. \right. \right. \right. \right.$$

$$\left. \left. \left. \left. \left. - \mathbb{1}_{Z_3 \in \left( \frac{\mathcal{X}-x-h(\alpha Z_1 + (1-\alpha) Z_2)}{h} \right)^c} \right) \right] \right] \mathbb{1}_{Z_1, Z_2 \in \frac{\mathcal{X}-x}{h}} \right|^{1+\delta} \right]$$

$$\leq \left( \frac{p_{\max}}{2\widetilde{\Lambda} \widetilde{p}_{\min}^{3/2}} \right)^{1+\delta} |\mathbb{E}_{Z_1, Z_2} \left( \|Z_1\| (\|Z_1 z_2\|) \right)|^{1+\delta} \mathbb{E}_X \left[ \|\nabla f(X)\|^{1+\delta} \left| \mathbb{E}_{Z_3} \left[ \frac{1}{2h} \|Z_3\| \left( \mathbb{1}_{Z_3 \in \left( \frac{\mathcal{X}-x}{h} \right)^c} \right) \right] \right|^{1+\delta} \right]$$

$$+ \left( \frac{p_{\max}}{2\widetilde{\Lambda}\widetilde{p}_{\min}^{3/2}} \right)^{1+\delta} \mathbb{E}_X \left[ \|\nabla f(X)\|^{1+\delta} \left| \mathbb{E}_{Z_1,Z_2} \left[ \|Z_1\| \|Z_1 - Z_2\| \right. \right. \right.$$

$$\left. \left. \left. \mathbb{E}_{Z_3} \left[ \frac{1}{2h} \|Z_3\| \mathbb{1}_{Z_3 \in \left( \frac{\mathcal{X} - x - h(\alpha Z_1 + (1-\alpha)Z_2)}{h} \right)^c} \right] \right] \mathbb{1}_{Z_1,Z_2 \in \frac{\mathcal{X}-x}{h}} \right|^{1+\delta} \right]$$

$$\leq T_{141} + T_{142} \,.$$

Bounding $T_{141}$ is again follows from similar calculation as we used to bound $\mathbb{E}\left[ |T_{11,n}|^{1+\delta} \right]$ and hence skipped. Now to bound $T_{142}$:

$$T_{142} = \left( \frac{p_{\max}}{2\widetilde{\Lambda}\widetilde{p}_{\min}^{3/2}} \right)^{1+\delta} \mathbb{E}_X \left[ \|\nabla f(X)\|^{1+\delta} \left| \mathbb{E}_{Z_1,Z_2} \left[ \|Z_1\| \|Z_1 - Z_2\| \right. \right. \right.$$

$$\left. \left. \left. \mathbb{E}_{Z_3} \left[ \frac{1}{2h} \|Z_3\| \mathbb{1}_{Z_3 \in \left( \frac{\mathcal{X} - x - h(\alpha Z_1 + (1-\alpha)Z_2)}{h} \right)^c} \right] \right] \mathbb{1}_{Z_1,Z_2 \in \frac{\mathcal{X}-x}{h}} \right|^{1+\delta} \right]$$

$$\leq \left( \frac{p_{\max}}{2\widetilde{\Lambda}\widetilde{p}_{\min}^{3/2}} \right)^{1+\delta} \mathbb{E}_X \left[ \|\nabla f(X)\|^{1+\delta} \mathbb{E}_{Z_1,Z_2} \left[ \|Z_1\|^{1+\delta} \|Z_1 - Z_2\|^{1+\delta} \right. \right.$$

$$\left. \left. \left| \mathbb{E}_{Z_3} \left[ \frac{1}{2h} \|Z_3\| \mathbb{1}_{Z_3 \in \left( \frac{\mathcal{X} - x - h(\alpha Z_1 + (1-\alpha)Z_2)}{h} \right)^c} \right] \right|^{1+\delta} \right] \mathbb{1}_{Z_1,Z_2 \in \frac{\mathcal{X}-x}{h}} \right]$$

$$\leq \left( \frac{p_{\max}}{2\widetilde{\Lambda}\widetilde{p}_{\min}^{3/2}} \right)^{1+\delta} \mathbb{E}_{X,Z_1,Z_2,Z_3} \left[ \|\nabla f(X)\|^{1+\delta} \|Z_1\|^{1+\delta} \|Z_1 - Z_2\|^{1+\delta} \right.$$

$$\left. \times \frac{1}{(2h)^{1+\delta}} \|Z_3\|^{1+\delta} \mathbb{1}_{X+h(Z_3+\alpha Z_1 + (1-\alpha)Z_2) \notin \mathcal{X}} \mathbb{1}_{Z_1,Z_2 \in \frac{\mathcal{X}-x}{h}, \|Z_1\| \vee \|Z_2\| \vee \|Z_3\| \leq \sqrt{\log \frac{1}{h}}} \right]$$

$$+ \left( \frac{p_{\max}}{2\widetilde{\Lambda}\widetilde{p}_{\min}^{3/2}} \right)^{1+\delta} \mathbb{E}_{X,Z_1,Z_2,Z_3} \left[ \|\nabla f(X)\|^{1+\delta} \|Z_1\|^{1+\delta} \|Z_1 - Z_2\|^{1+\delta} \right.$$

$$\left. \times \frac{1}{(2h)^{1+\delta}} \|Z_3\|^{1+\delta} \mathbb{1}_{X+h(Z_3+\alpha Z_1 + (1-\alpha)Z_2) \notin \mathcal{X}} \mathbb{1}_{Z_1,Z_2 \in \frac{\mathcal{X}-x}{h}, \|Z_1\| \vee \|Z_2\| \vee \|Z_3\| > \sqrt{\log \frac{1}{h}}} \right]$$

Now it is bounded via similar argument used to bound $\mathbb{E}[|T_{11,n}|^{1+\delta}]$ and hence skipped. Therefore we have established all the terms in the bound of $g_{h_n}(x)$ in equation (B.9) is uniformly integrable which further implies that the sequence of functions $\{g_{h_n}(x)\}$ is uniformly integrable, which concludes the proof.

## B.2 Proof of Theorem 3.1

*Proof.* The penalized objective using KL divergence on the probability space for unnormalized graph Laplacian can be written as:

$$g(y_1,\ldots,y_n) = \sum_{i=1}^{n} \mathsf{KL}\left( P_{y_i} || P_{\hat{y}_i} \right) + \frac{\lambda}{2} \sum_{i \neq j} W_{ij} \mathsf{KL}\left( P_{y_i} || P_{y_j} \right) \,.$$

where $y$ is the matrix with rows being $y_1, \ldots, y_n$. Note that the probability vector $y_i$ can be written as:

$$y_i = \left[ \frac{e^{o_{i1}}}{\sum_{j=1}^{k} e^{o_{ij}}}, \frac{e^{o_{i2}}}{\sum_{j=1}^{k} e^{o_{ij}}}, \ldots, \frac{e^{o_{ik}}}{\sum_{j=1}^{k} e^{o_{ij}}} \right]$$

with $o_i$ being the output of the penultimate layer of the neural network and $P_{y_i}$ is the k-class multinomial distribution with probabilities specified by $y_i$. For the rest of the analysis, define $\eta_i$ (resp. $\hat{\eta}_i$) $\in \mathbb{R}^{K-1}$ to be the natural parameter corresponding to $y_i$, i.e., $\eta_{ij} = \log \left( y_{ij}/y_{ik} \right) = o_{ij} - o_{i,k}$. The multinomial p.m.f. is of the form:

$$f_{y_i}(\mathbf{x}) = \Pi_{j=1}^{k} y_{ij}^{x_j} = e^{\sum_{j=1}^{k-1} x_j \eta_{ij} - \log\left(1 + \sum_{j=1}^{k-1} e^{\eta_{ij}}\right)} := e^{\sum_{j=1}^{k-1} x_j \eta_{ij} - A(\eta_i)}$$

Also, from the properties of the distributions from exponential family, we know $\mathbb{E}_{X \sim P_{y_i}}[X] = \nabla A(\eta_i)$. For any $i, j$ the KL divergence between $P_{y_i}$ and $P_{y_j}$ is:

$$
\begin{aligned}
KL\left(P_{y_i} || P_{y_j}\right) &= \int_{\mathcal{X}} \log \frac{f_{y_i}(x)}{f_{y_j}(x)} f_{y_i}(x)\, d\mu(x) \\
&= \int_{\mathcal{X}} \left\{ (\eta_i - \eta_j)^{\top} x - (A(\eta_i) - A(\eta_j)) \right\}\, f_{y_i}(x)\, d\mu(x) \\
&= (\eta_i - \eta_j)^{\top} \mathbb{E}_{X \sim P_{y_i}}[X] - (A(\eta_i) - A(\eta_j)) \\
&= (\eta_i - \eta_j)^{\top} \nabla A(\eta_i) - (A(\eta_i) - A(\eta_j)) \\
&= d_A\left(\eta_j, \eta_i\right) .
\end{aligned}
$$

where $d_A$ is the Bregman divergence with respect to $A$ and $\mu$ is the counting measure as we are dealing with discrete random variable. Now consider the case when we want to minimize the following objective function:

$$
\hat{\theta} = \arg\min_{\theta} \sum_{i=1}^{n} \omega_i KL(P_\theta || P_{\theta_i})
$$

We can minimize above *barycenter* problem with respect to the natural parameters and then transform it back to the original parameter. To be precise, first we solve:

$$
\hat{\eta} = \arg\min_{\eta} \sum_{i=1}^{n} \omega_i\, d_A\left(\eta_i, \eta\right)
$$

then transform $\hat{\eta}$ to $\hat{\theta}$. Then $\hat{\eta}$ satisfies the following first order condition:

$$
\left. \frac{d}{d\eta} \sum_{i=1}^{n} \omega_i\, d_A\left(\eta_i, \eta\right) \right|_{\eta=\hat{\eta}} = 0
$$

$$
\implies \left. \frac{d}{d\eta} \sum_{i=1}^{n} \omega_i \left[ A(\eta_i) - A(\eta) - \nabla A(\eta)^{\top} (\eta_i - \eta) \right] \right|_{\eta=\hat{\eta}} = 0
$$

$$
\implies \sum_{i=1}^{n} \omega_i \left[ -\nabla A(\hat{\eta}) + \nabla A(\hat{\eta}) - \nabla^2 A(\hat{\eta})\, (\eta_i - \hat{\eta}) \right] = 0
$$

$$
\implies -\nabla^2 A(\hat{\eta}) \sum_{i=1}^{n} \omega_i\, (\eta_i - \hat{\eta}) = 0
$$

$$
\implies \sum_{i=1}^{n} \omega_i\, (\eta_i - \hat{\eta}) = 0 \implies \hat{\eta} = \left( \frac{\sum_{i=1}^{n} \omega_i \eta_i}{\sum_{i=1}^{n} \omega_i} \right) \qquad [\because \nabla^2 A(\eta) \succ 0 \text{ on the domain}] .
$$

The same optimal solution can be found via minimizing the following quadratic problem:

$$
\hat{\eta} = \arg\min_{\eta} \frac{1}{2} \sum_{i=1}^{n} \omega_i\, \|\eta - \eta_i\|^2
$$

Hence for each i, fixing $y_j$ for $j \neq i$ our update step is:

$$
\tilde{\eta}_i \leftarrow \arg\min_{\eta} \left[ \|\eta - \hat{\eta}_i\|^2 + \sum_{j \neq i} W_{ij} \|\eta - \eta_j\|^2 \right]
$$

$$
y_i \leftarrow \left[ \frac{e^{\tilde{\eta}_{i1}}}{1 + \sum_{j=1}^{K} e^{\tilde{\eta}_{ik}}}, \ldots, \frac{e^{\tilde{\eta}_{iK-1}}}{1 + \sum_{j=1}^{K} e^{\tilde{\eta}_{ik}}}, \frac{1}{1 + \sum_{j=1}^{K} e^{\tilde{\eta}_{ik}}} \right] .
$$

$\square$

## B.3 Extension of Theorem 3.1

The following theorem extends the result of Theorem 3.1 to general Bregman divergence function:

**Theorem B.4.** *Suppose $\tilde{y}_i$ is the minimizer of the following objective function:*

$$\tilde{y}_i = \arg\min_y \left\{ D_F(y, \tilde{y}_i) + \sum_{j \neq i} W_{i,j} D_F(y, \tilde{y}_j) \right\} .$$

*Then $\tilde{y}_i$ is also minimizer of the following squared error loss:*

$$\tilde{y}_i = \arg\min_y \left[ \|y - \hat{y}_i\|^2 + \sum_{j \neq i} W_{ij} \|y - \tilde{y}_j\|^2 \right]$$

*Proof.* The proof of quite similar to that of Theorem 3.1. Recall that for a convex function $F$, the Bregman divergence is defined as:

$$D_F(x, y) = F(x) - F(y) - \langle x - y, \nabla F(y) \rangle .$$

KL divergence is a special case of Bregman divergence when $F(x) = \sum_i x_i \log x_i$. As it is evident from the proof of Theorem 3.1, minimizing KL divergence becomes equivalent to minimizing the weighted combination of Bregman divergence with respect to log partition function $A$. Now consider a general form equation (3.8):

$$(\tilde{y}_1, \ldots, \tilde{y}_n) = \arg\min_{y_1, \ldots, y_n} \left[ \sum_{i=1}^n \left\{ D_F(y_i, \hat{y}_i) + \sum_{j \neq i} W_{i,j} D_F(y_i, y_j) \right\} \right]$$

In our co-ordinate descent algorithm, for a fixed $i$ we solve:

$$\tilde{y}_i = \arg\min_y \left\{ D_F(y, \hat{y}_i) + \sum_{j \neq i} W_{i,j} D_F(y, \tilde{y}_j) \right\}$$

$$\triangleq \arg\min_y \left\{ \sum_{j=1}^n \omega_j(y, z_j) \right\}$$

where $z_j = \tilde{y}_j$ for $j \neq i$ and $z_i = \hat{y}_i$, and for the weights $\omega_j = W_{i,j}$ for $j \neq i$, $\omega_i = 1$. It follows via similar calculation as in the proof of Theorem 3.1 that:

$$\tilde{y}_i = \frac{\sum_j \omega_j z_j}{\sum_j \omega_j}$$

which, as argued before is the solution of the following quadratic optimization problem:

$$\tilde{y}_i = \arg\min_y \left[ \|y - \hat{y}_i\|^2 + \sum_{j \neq i} W_{ij} \|y - \tilde{y}_j\|^2 \right] .$$

This completes the proof. $\qquad\square$