# OpenReview forum: "Post-processing for Individual Fairness"
_NeurIPS.cc/2021/Conference — NeurIPS 2021 Poster_

### Official Review · Reviewer_63Qx · 2021-07-16

**Rating:** 7
**Confidence:** 3

**Summary:**

This paper attempts to instantiate the individual fairness solution concept of Dwork et al in a manner which relaxes some of the stringency of a global Lipschitz constraint. The proposed formulation applies a graph Laplacian smoothing technique to post-process the predictions of an arbitrary model. The graph is chosen so that edge weights decay exponentially with decreasing similarity (wrt the "fairness" metric), with a cutoff to 0 at some point. Under some assumptions, this regularizer is shown to enforce a much more local version of individual fairness. Experimental results show that this results in a better tradeoff with accuracy than the original notion, and similar or better performance compared to a recently proposed method.


**Limitations And Societal Impact:**

yes

**Main Review:**

The main strength of this paper is to propose an alternate way of instantiating individual fairness which is easy to apply and which appears to provide a better tradeoff between fairness and accuracy. The Laplacian method is well-motivated in terms of a connection to a local notion of fairness. This method does not address the (often prohibitive) challenge of identifying a suitable metric with which to define individual fairness. However, given that such a method can be identified, it proposes a nice way of enforcing it.

A few questions:

(1) What is the impact of the parameters theta and tau on the behavior of the method?

(2) What is the conceptual relation of the Laplacian method to the alternate definition of individual fairness proposed in [36]?

(3) How does the method relate to alternate proposals to use a graph structure to instantiate individual fairness such as http://www.vldb.org/pvldb/vol13/p506-lahoti.pdf or https://dl.acm.org/doi/abs/10.1145/3394486.3403080?


**Time Spent Reviewing:**

1.5

---

> ### Author Response · Authors · 2021-08-09
> **Response**
>
>
> Thank you for the feedback and questions. Please see our responses below.
>
> **What is the impact of the parameters theta and tau on the behavior of the method?**
>
> As long as $\theta$ is sufficiently small, the method is rather insensitive to $\theta$ (and the corresponding graph is close to being binary). We didn't find $\theta$ to be particularly important and used a small value in all experiments. Parameter $\tau$ is more interesting -- smaller $\tau$ leads to a sparser similarity graph with edges only between the most similar individuals, i.e., the individual fairness that we are enforcing is becoming "more local". In other words, reducing $\tau$ reduces the amount of individual fairness, but improves accuracy. To explore the fairness-accuracy trade-offs (see Figures 2 and 3) we varied $\tau$. For the results in Tables 1 and 2 we selected hyperparameters based on a validation portion of the data. Please see Section 2.3 of the supplement for the additional details regarding the hyperparameters.
>
> **What is the conceptual relation of the Laplacian method to the alternate definition of individual fairness proposed in [36]?**
>
> The two notions can be made similar by picking the $\epsilon$ and $\delta$ parameters in the DIF [36] appropriately. However, they are not identical. DIF is a more stringent definition of IF because it precludes the existence of *any* point $x'$ in an $\epsilon$ neighborhood of $x$ w.r.t. the fair metric such that $d_{\mathcal{Y}}(h(x),h(x')) > \delta$. On the other hand, the Laplacian regularizer only precludes the existence of another $x'$ in the dataset such that $d_{\mathcal{Y}}(h(x),h(x')) > \delta$. Enforcing DIF, i.e., the in-processing method of [36], requires solving the inner optimization problem (an adversarial attack to find the worst case $x'$), which is a lot more intensive computationally.
>
> **How does the method relate to alternate proposals to use a graph structure to instantiate individual fairness such as [1, 2]?**
>
> We thank the reviewer for pointing out additional related works, and we will cite them in the revised version.
>
> The main differences between our work and [1] and [2] are (i) the ML tasks we consider and (ii) [1] and [2] provide no theoretical justification that their methods actually enforce IF. Regarding ML tasks, [1] considers representation learning, while [2] considers graph mining tasks, such as clustering, representation learning, and ranking. We consider post-processing the outputs of models for supervised learning. In particular, our experiments show that our method is successful in debiasing large language models on Bios and Toxicity datasets.
>
> Regarding theory, we show that graph Laplacian regularizers do NOT enforce Dwork et al's original definition of IF. Rather, depending on the choice of the Laplacian matrix, graph Laplacian regularizers enforce local notions of IF (see Thm 4.6). In particular, picking a random-walk Laplacian enforces local IF (see Def 4.1), and picking the unnormalized Laplacian enforces a density weighted version of our definition of local IF (see the first conclusion of Thm 4.6).
> This theoretical insight is important: as we state in the conclusion (lines 348-350), some applications may require global IF in which case Laplacian-based methods should not be used, and in lines 298-304 we discuss practical implications of different Laplacian formulations, e.g., unnormalized Laplacian weighs higher density regions higher which could be good or bad depending on the context. The practitioner needs to be aware of all the trade-offs that our theory demonstrates when choosing a method for a specific application. We emphasize that [1,2] only considered unnormalized Laplacian without investigating the corresponding individual fairness guarantees.
>
>
> [1] Lahoti, P., Gummadi, K. P., & Weikum, G. (2019). Operationalizing individual fairness with pairwise fair representations.
>
> [2] Kang, J., He, J., Maciejewski, R., & Tong, H. (2020). Inform: Individual fairness on graph mining.

---

### Official Review · Reviewer_bTqy · 2021-07-16

**Rating:** 5
**Confidence:** 3

**Summary:**

This paper proposes a method for post-processing a model for guaranteeing individual fairness properties based on graph Laplacian regularization. Their main result is that this method achieves a weakened definition of individual fairness (under certain assumptions). The authors also evaluate this method on text datasets for fine-tuned BERT models.

The method takes as input a graph consists of pairs which are close according to the similarity metric and solves an optimization problem using a graph Laplacian quadratic form regularizer to achieve fairness conditions. The fairness guarantee is that in expectation, the function is Lipschitz on sufficiently small distances (a sufficient condition for Mahalanobis distance is that the derivative is bounded in expectation).


**Limitations And Societal Impact:**

The authors adequately discuss the societal impact of this work.

**Main Review:**

The authors describe two problems with the classical post-processing approach (Dwork et al): (1) it is computationally inefficient since it requires solving an optimization problem with O(n^2) constraints, and (2) global individual fairness can be in tension with accuracy. The idea of designing a post-processing method that achieves more desirable guarantees in practice is well-motivated and interesting. The paper is generally well-written.

Although the main result (Theorem 4.6) appears technically sound, it does not seem particularly surprising that the graph Laplacian approach guarantees local individual fairness. In particular, the graph Laplacian regularization term can be written as the sum of the squares of distances between pairs weighted appropriately by the similarity metric distance (as shown in equation 3.2).

Another weakness is that the authors provide a limited comparison of local individual fairness definition and the typical individual fairness definition. First, the empirical results in Section 5.1 are fairly limited and do not offer a thorough the fairness-accuracy tradeoffs for these two definitions. Second, since the local individual fairness is permitted to violate the Lipschitz constraint for large distances, it would be natural to also compare with a global individual fairness with respect to an adapted metric d’ where d’(u, v) = $\infty$ if d(u,v) > \tau and d’(u,v) = d(u,v)  otherwise.

-----------
Update: Thanks to the authors for their detailed responses (and for correcting the mistake in my definition of $d'$). However, I am still not entirely convinced by the proposed fairness definition, which seems to be a significant relaxation of IF. Thus, I have decided to maintain my score.


**Time Spent Reviewing:**

4

---

> ### Author Response · Authors · 2021-08-09
> **Response**
>
>
> We thank the reviewer for the comments. Please find our responses below.
>
> **Although the main result (Theorem 4.6) appears technically sound, it does not seem particularly surprising that the graph Laplacian approach guarantees local individual fairness. In particular, the graph Laplacian regularization term can be written as the sum of the squares of distances between pairs weighted appropriately by the similarity metric distance (as shown in equation 3.2).**
>
> While we agree that connection between Laplacian regularization and individual fairness is intuitive, we argue that the precise relation demonstrated by our theory and experiments is substantially more insightful than the basic intuition. First, Laplacian regularization does not actually enforce Dwork et al's original definition of IF. We introduce the new notion of local IF and show that a specific version of graph Laplacian, i.e. the normalized random-walk Laplacian, is required to achieve it. The more common unnormalized graph Laplacian (eq 3.2) yields a density weighted variation of local IF (see the first conclusion of Thm 4.6). This theoretical insight is important: as we state in the conclusion (lines 348-350), some applications may require global IF in which case Laplacian based methods should not be used, and in lines 298-304 we discuss practical implications of different Laplacian formulations, e.g. unnormalized Laplacian weighs higher density regions higher which could be good or bad depending on the context. Practitioner needs to be aware of all the trade-offs that our theory demonstrates when choosing a method for a specific application. Deploying methods for achieving fairness based on intuition and without in-depth understanding may cause new unforeseen biases.
>
> **Another weakness is that the authors provide a limited comparison of local individual fairness definition and the typical individual fairness definition. First, the empirical results in Section 5.1 are fairly limited and do not offer a thorough the fairness-accuracy trade-offs for these two definitions.**
>
> If the reviewer has specific questions regarding differences between local and classical (global) IF, we would be happy to clarify. Presently, there is an entire section (Section 5.1) of the paper devoted to discussing and illustrating differences between local IF (correspondingly GLIF and GLIF-NRW) and global IF (correspondingly IF-constraints).
> In Figure 2 (left) we show individual fairness and accuracy trade-off. We also present results for group fairness and accuracy trade-off in Figure 2 (center). We find that our methods for enforcing local IF always outperform the global IF constraints method in terms of these trade-offs and discuss the results in lines 275-282. Next we discuss the reasons for improved performance of the local IF by investigating the types of constraints that local IF violates: in Figure 2 (right) we present a (normalized) histogram for the frequencies of violations of the IF constraints. While there are quite a lot of violations of the IF constraints overall (blue), they are for words other than names.
> This is because the fair distances between names (orange) are small, while the violations only occur at large distance. There are almost no IF violations for names (green, and barely visible). This demonstrates the notion of local IF, where violations for large fair distances do not matter as much and the focus lies on those elements which are similar. We discuss this result in more details in lines 283-297.
>
>
> **Second, since the local individual fairness is permitted to violate the Lipschitz constraint for large distances, it would be natural to also compare with a global individual fairness with respect to an adapted metric d’ where $d'(u, v) = 1 if d(u,v) > \tau$ and $d'(u,v) = d(u,v)$ otherwise.**
>
> We are not sure about the intuition behind the adapted metric $d'$. If we truncate some fair metric and try to enforce global IF w.r.t. it, then the conditions become even more stringent: for any two points $u, v$ which were far away from each other (i.e. when $d'(x, y) = 1$), we now need $d_\mathcal{Y}(h(u), h(v)) \le L$ (where $d_\mathcal{Y}$ is the distance metric on the output space), which is even stronger than the condition $d_\mathcal{Y}(h(u), h(v)) \le L d(u, v)$. The idea of local IF is completely opposite: ignore the constraints when $d(u, v)$ is large. If instead we set $d(u, v)$ to be $\infty$ when it is bigger than $\tau$, then yes, the problem becomes similar to local IF. However, it is still a convex optimization problem with large number of constraints and solving it directly (as opposed to using our method) will be computationally inefficient.
>
> Please respond to our post to let us know if the clarifications above suitably address your concerns about our work.  We are happy to address any remaining points during the discussion phase; if the responses above are sufficient, we kindly ask that you consider raising your score.

---

### Official Review · Reviewer_9cRG · 2021-07-17

**Rating:** 6
**Confidence:** 3

**Summary:**

The authors proposed a post-processing algorithm for individual fairness. The algorithm get access of the predictions of the original model and a similarity graph between individuals, and solves the IF post-processing problem as a graph smoothing problem. The motivation to reduce bias of fine-tuned model is interesting. The paper is well written. However, I have some concerns about the effectiveness of the proposed method.

In Figure 1, the node signals of Alice and Bob are different, cross for Alice and tick for Bob. After connecting the node Alice and node Bob which enjoy the same qualities, their signals should be the same. I wonder what the signal should be. The experimental results on "Table 1: Results for the Bios task" and "Table 2: Results for the Toxicity task" show that the difference of the proposed method and the baseline is trivial is in terms of test accuracy and the prediction consistency. But if there are bias of the algorithm's prediction as shown in Figure 1, the difference between the test accuracy of the proposed method and the baseline should be significant. If the algorithm is biased but the proposed method in this work does not connect similar nodes in the graph, it tends to yield similar predictions as the baseline.

In line 72, the experiments are conducted to reduce bias of fine-tuned Bert. Is there any assumption that the fine-tuned Bert is biased?

**Limitations And Societal Impact:**

See above.

**Main Review:**

See above.

**Time Spent Reviewing:**

2

---

> ### Author Response · Authors · 2021-08-09
> **Response**
>
>
> Thank you for the review. We address questions and concerns below.
>
> **Alice and Bob are different. I wonder what the signal should be.**
>
> To satisfy individual fairness, the signal in the diagram should be the same for Alice and Bob because they have the same qualification. Whether they should be displayed the job ad, should depend on whether the job requires knowledge of the programming language Ruby (and not on name, gender, etc.). Note that individual fairness (i.e., eq. (2.1)) does not concern the true label, it only ensures similar individuals should be treated similarly.
>
> **Difference of the proposed method and the baseline is trivial is in terms of test accuracy and the prediction consistency.**
>
> The goal of our method is to improve individual fairness while maintaining the original test accuracy, so the test accuracy should be similar for the baseline and our method. On the other hand, the improvement of prediction consistency (i.e., individual fairness) is from 94.2\% to 98.8\% on Bios and from **61.4\%** to **84.4\%** on Toxicity. This improvement is not trivial.
>
> **But if there are bias of the algorithm's prediction as shown in Figure 1, the difference between the test accuracy of the proposed method and the baseline should be significant. If the algorithm is biased but the proposed method in this work does not connect similar nodes in the graph, it tends to yield similar predictions as the baseline.**
>
> Similar values of test accuracy do not imply similar predictions (unless the accuracy is 100\%). Accuracy is an aggregate measure over the whole test set: it is possible to make mistakes on different test points and to have exactly identical accuracy. Consider this example: in the Toxicity dataset, many comments with the word "gay" in them are labeled as toxic. Fine-tuned BERT learns to associate word "gay" with toxicity which does help with the test accuracy in this case, however it is not the way we want to achieve accuracy, i.e. it is an instance of algorithmic bias. The goal of our method is to preserve accuracy while making sure the model does not rely on identity words such as "gay" to make its prediction, which is measured by prediction consistency. Our method does achieve substantial improvements on prediction consistency. How does it manage to maintain similar accuracy as the baseline? The algorithm may make new mistakes on a small amount of comments with word "gay" that are labeled as toxic since it is harder to classify them without using this spurious correlation, but it might avoid mistakes on comments with word "gay" that are labeled as non-toxic since it is not biased by the spurious correlation.
>
> **In line 72, the experiments are conducted to reduce bias of fine-tuned Bert. Is there any assumption that the fine-tuned Bert is biased?**
>
> Biases in large language models, including those based on fine-tuning BERT, have been demonstrated in many prior works, e.g. see references [3, 23, 28] in the paper. Same is true in our experiments as can be seen by low prediction consistency values of the baseline, which indicates biases. Biases in BERT and models obtained by fine-tuning it are rather a fact, than an assumption.

---

### Official Review · Reviewer_LaWU · 2021-07-19

**Rating:** 4
**Confidence:** 3

**Summary:**

This paper proposes a post-processing approach to promote individual fairness of the outcome of any machine learning model. Based on the problem formulation, a basic approach together with several variants utilizing graph Laplacian to enforce individual fairness (GLIF) are presented. In the experiments, GLIF shows better efficiency, fairness metrics, and balance between test accuracy and prediction consistency. Generally, this paper is with limited technical novelty and lacks baselines for a comprehensive comparison. However, making trails to enforce individual fairness in different realms, such as debiasing the output of language models, is always encouraged.

**Limitations And Societal Impact:**

Did not see any potential negative societal impact.

**Main Review:**

Strengths:
(1) This paper considers a practical scenario and is well-organized.
(2) This paper is well-motivated, and utilizing graph Laplacian to promote individual fairness is a reasonable way despite the lack of novelty.
(3) Compared to the chosen baselines, the proposed GLIF achieves better performance in different ways.

Weaknesses:
(1) Some techniques proposed in this paper lack novelty and motivation.
(2) This paper lacks the study on related work, which should have been included to enrich the baselines for a comprehensive investigation.
(3) Some potential limitations are not well considered and presented.

Details:
In this paper, a post-processing approach based on graph Laplacian to promote individual fairness is proposed. On the one hand, this paper considers a practical scenario, which is the post-processing stage, to promote individual fairness. Generally, this paper is well-motivated and well-organized. Experiments also verify GLIF’s better performance on efficiency, fairness metrics, and balance between test accuracy and prediction consistency. Nevertheless, on the other hand, the drawbacks of this paper are also obvious. Firstly, some techniques proposed in this paper lacks novelty. For example, this paper lacks a survey on related works, which should have been included to notify the audience that similar approaches have been proposed and adopted in various previous works, such as [1, 2]. Also, although enforcing individual fairness based on graph Laplacian is already a widely adopted approach, what would be the specific motivation in this paper? Secondly, some techniques in this paper also lack motivation. For example, what would be the advantage of making use of alternative discrepancy measures on the output space compared with your approach in section 3.2.1? Both of the outputs here can be regarded as a distribution. Also, if KL Divergence is adopted here, do we need to explicitly assign the distribution? If the answer is positive, would you believe that this can also be a limitation of the approach in section 3.2.4? Can you provide a theoretical analysis on what is the main motivation and advantage of the alternative discrepancy measure? Thirdly, some potential limitation is not well considered. Beyond the limitation we mentioned above, why should we base on sensitive attributes to promote individual fairness in Example 4.2? What if there is no sensitive attribute assignment? What is the definition of the mentioned sensitive attribute here? Also, does the dispersion matrix only have the ability to reflect linear relationships? Will this be another limitation? Beyond the problems I mentioned above, I would still have the following concerns:
(1) For inductive settings, only performing a single coordinate descent step could be far from optimal, and the step length can be hard to specify. How could you tackle such problem?
(2) What is the exact definition of the bandwidth here? In the paper, is $h$ a function or a bandwidth parameter?

[1] Lahoti, P., Gummadi, K. P., & Weikum, G. (2019). Operationalizing individual fairness with pairwise fair representations. arXiv preprint arXiv:1907.01439.

[2] Kang, J., He, J., Maciejewski, R., & Tong, H. (2020, August). Inform: Individual fairness on graph mining. In Proceedings of the 26th ACM SIGKDD International Conference on Knowledge Discovery & Data Mining (pp. 379-389).


**Time Spent Reviewing:**

3.5 hours.

---

> ### Author Response · Authors · 2021-08-09
> **Response**
>
> We thank the reviewer for the comments. We address questions and concerns below.
>
> **Firstly, some techniques proposed in this paper lacks novelty. For example, this paper lacks a survey on related works, which should have been included to notify the audience that similar approaches have been proposed and adopted in various previous works, such as [1, 2].**
>
> We thank the reviewer for pointing out additional related works, and we will cite them in a revised version.
>
> The main differences between our work and [1] and [2] are (i) the ML tasks we consider and (ii) [1] and [2] provide no theoretical justification that their methods actually enforce IF. Regarding ML tasks, [1] considers representation learning, while [2] considers graph mining tasks, such as clustering, representation learning, and ranking. We consider post-processing the outputs of models for supervised learning. In particular, our experiments show that our method is successful in debiasing large language models on Bios and Toxicity datasets.
>
> Regarding theory, we show that graph Laplacian regularizers do NOT enforce Dwork et al's original definition of IF. Rather, depending on the choice of the Laplacian matrix, graph Laplacian regularizers enforce local notions of IF (see Thm 4.6). In particular, picking a random-walk Laplacian enforces local IF (see Def 4.1), and picking the unnormalized Laplacian enforces a density weighted version of our definition of local IF (see the first conclusion of Thm 4.6).
> This theoretical insight is important: as we state in the conclusion (lines 348-350), some applications may require global IF in which case Laplacian-based methods should not be used, and in lines 298-304 we discuss practical implications of different Laplacian formulations, e.g., unnormalized Laplacian weighs higher density regions higher which could be good or bad depending on the context. The practitioner needs to be aware of all the trade-offs that our theory demonstrates when choosing a method for a specific application. We emphasize that [1,2] only considered unnormalized Laplacian without investigating the corresponding individual fairness guarantees.
>
> **What would be the advantage of making use of alternative discrepancy measures on the output space compared with your approach in section 3.2.1? Both of the outputs here can be regarded as a distribution. Also, if KL Divergence is adopted here, do we need to explicitly assign the distribution? If the answer is positive, would you believe that this can also be a limitation of the approach in section 3.2.4? Can you provide a theoretical analysis on what is the main motivation and advantage of the alternative discrepancy measure?**
>
> Recall that individual fairness definition (eq. (2.1)) requires a metric on the model outputs $d_{\mathcal{Y}}$. Alternative discrepancy measures studied in section 3.2.4 allow a practitioner to choose different output metrics when using our method. Choosing output discrepancy measures is similar to choosing loss functions in supervised learning (loss is a discrepancy measure between model outputs and the truth), e.g. in regression MSE loss ($d_{\mathcal{Y}}$ is a squared Euclidean distance) is typically preferred, while in classification, cross entropy is a common choice (although MSE loss is also an option, it leads to poor calibration). Cross entropy loss corresponds to $d_{\mathcal{Y}}$ being a KL divergence. To see this, recall that in classification models typically output probability distributions, e.g. a neural network trained with cross-entropy loss outputs a distribution over classes. There is no need to "explicitly assign" any distributions. Cross entropy and KL divergence are identical up to an additive constant (CE = $-\sum_k y_k \log \hat y_k = \text{KL}(y||\hat y) + \sum_k y_k \log y_k$), i.e. minimizing cross entropy is the same as minimizing KL divergence. To summarize, the advantage of allowing different output discrepancy measures is the same as the advantage of supporting different loss functions such as those typical in regression and classification problems. This is not a limitation. The theoretical shortcomings of MSE loss in classification is a standard result in the literature.
>
>
> **Why should we base on sensitive attributes to promote individual fairness in Example 4.2? What if there is no sensitive attribute assignment? What is the definition of the mentioned sensitive attribute here?**
>
> We would like to clarify that our method does not require any sensitive attributes. Example 4.2 uses a fair metric proposed in the prior work (which does use sensitive attributes, but it is independent of our method) to illustrate the difference between global and local individual fairness and to show that some existing fair metrics satisfy our definition of local IF. Our method can be used with *any* fair metric, regardless of whether it depends on sensitive attributes or not. In fact, in our experiments, we didn't use any sensitive attributes. We will clarify the example to emphasize that our approach is more general and does not rely on sensitive attributes.
>
>
> **For inductive settings, only performing a single coordinate descent step could be far from optimal, and the step length can be hard to specify. How could you tackle such problem?**
>
> The coordinate update step for the inductive setting is a closed form optimal solution for the incoming data point holding the solution for previously seen points fixed -- see Equation (3.7). We emphasize that it is a closed form solution; hence no step length is involved.
>
>
> **What is the exact definition of the bandwidth here? In the paper, is h a function or a bandwidth parameter?**
>
> We thank the reviewer for noting the notation overload. In Equation 2.1 $h$ is a function. In the center of page 6, $h$ is a bandwidth parameter, which is a reparameterized version of $\theta$ (see the paragraph between lines 213 and 214). This is a scale parameter used in the analysis of the graph Laplacian. We will change the bandwidth parameter notation.
>
> Please respond to our post to let us know if the clarifications above suitably address your concerns about our work.  We are happy to address any remaining points during the discussion phase; if the responses above are sufficient, we kindly ask that you consider raising your score.

---

### Decision · Program_Chairs · 2021-09-28

**Decision:**

Accept (Poster)

**Comment:**

Thank you for your submission. This paper gives a new post-processing technique for obtaining a relaxation of the individual fairness criterion. While the paper contributes to the literature on individual fairness, we felt that there was a lack of motivation or justification of why such local relaxation of individual fairness is a meaningful criterion. In particular, it seems that the relaxed individual fairness notion was proposed since it is satisfied by the Laplacian method. Thus, the paper can be strengthened if the authors could provide more exposition around their definition.

**Consistency Experiment:**

NeurIPS has a long history of experimentation. In 2014, NeurIPS ran an experiment in which 10% of submissions were reviewed by two independent committees to quantify the randomness in the review process. This year, we repeated a variant of this experiment to see how the quality of the review process has changed over time.  This paper was part of the experiment and was therefore assigned to two committees (consisting of reviewers, an Area Chair, and a Senior Area Chair) that reached independent decisions.  If both committees made the same recommendation, this recommendation was followed. If a single committee recommended acceptance, the paper was accepted (with the exception of a few cases in which the other committee identified what we considered a fatal flaw, e.g., an error in a key result).

This copy’s committee reached the following decision: **Reject**

The other committee assigned to the paper recommended **Accept (Poster)**.  You can find the other set of reviews, along with any follow up discussion with the authors here:
https://openreview.net/forum?id=qGeqg4_hA2